# Contrastive Flow Map Matching

**Junyu Zhang**[1]  **Daochang Liu**[2]  **Younghyun Kim**[3]  **Jong Hwan Ko**[1]  **Shichao Zhang**[4]  **Chang Xu**[5]
**Eunbyung Park**[3]

## Abstract

Flow map matching (FMM) enables one- and few-step sampling for diffusion-style generation, yet its performance is often hindered by the mismatch between ground-truth training transitions and model-induced flow maps. We propose **Contrastive Flow Map Matching (CFMM)**, a principled framework that explicitly aligns FMM training with practical sampling. Our approach is motivated by a joint-KL decomposition on the reverse KL divergence, which decomposes the distributional gap into a marginal mismatch over intermediate states and a conditional mismatch in endpoint reconstruction. This analysis motivates two complementary objectives: average-velocity regression for marginal alignment and a sampling-aligned InfoNCE contrastive loss for conditional refinement. CFMM is a training-only plug-in for pre-trained FMMs, incurs no inference-time overhead, and supports training FMMs from scratch. Experiments on CIFAR-10, ImageNet, and LSUN across multiple FMM baselines demonstrate consistent improvements in fidelity and perceptual quality with only modest additional training cost.

## 1. Introduction

High-quality generative modeling has become a cornerstone of modern machine learning, enabling rapid progress in generating images (Ma et al., 2024; Yu et al., 2025), videos (Bar-Tal et al., 2024), speech (Jia et al., 2025), audio (Huang et al., 2023), and 3D scenes (Sun et al., 2025; Wu et al., 2025b). Among the most successful approaches, diffusion (Karras

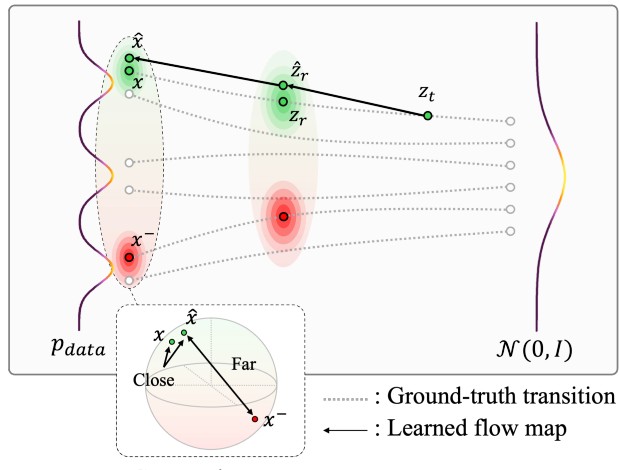

*Figure 1.* **Contrastive Flow Map Matching**. FMMs learn finite-time transport maps that move samples from a Gaussian prior $\mathcal{N}(0, I)$ (right) toward the data distribution $p_{\text{data}}$ (left) via large transitions along the sampling trajectory. Starting from a noisy state $z_t$, an FMM predicts an intermediate state $\hat{z}_r$ and then maps it to a final sample $\hat{x}$. Due to imperfect average-velocity estimation, the model-induced intermediate distribution may deviate from the ground-truth bridge state $z_r$ (marginal mismatch), which subsequently degrades the endpoint mapping from $\hat{z}_r$ to $\hat{x}$ (conditional mismatch). CFMM addresses this training–sampling discrepancy by combining average-velocity regression to improve transport dynamics with a sampling-aligned contrastive objective that pulls matched pairs $(\hat{x}, x)$ close in feature space while pushing negatives $x^-$ far apart, thereby improving semantic fidelity and fast-sampling quality without changing inference-time sampling.

et al., 2022; Rombach et al., 2022) and flow-based (Liu et al., 2023; Albergo & Vanden-Eijnden, 2023) generative models have demonstrated exceptional fidelity and mode coverage, benefiting from stable likelihood-free training objectives and strong scalability. However, these gains often come with a substantial inference cost: many models rely on iterative sampling that requires dozens to hundreds of function evaluations per sample (Song et al., 2021; Lu et al., 2022; Lipman et al., 2023). This computational burden poses a major challenge for latency-critical deployments, including mobile or edge generation, interactive content creation, and real-time decision-making (Xie et al., 2025).

The pursuit of *fastforward* generative models—producing

---

[1]Department of Electrical and Computer Engineering, Sungkyunkwan University. [2]School of Physics, Mathematics and Computing, University of Western Australia. [3]Department of Artificial Intelligence, Yonsei University. [4]Guangxi Key Lab of Multi-Source Information Mining & Security, Guangxi Normal University. [5]School of Computer Science, University of Sydney. Correspondence to: Eunbyung Park <epark@yonsei.ac.kr>.

*Proceedings of the 43$^{rd}$ International Conference on Machine Learning*, Seoul, South Korea. PMLR 306, 2026. Copyright 2026 by the author(s).

high-quality samples in only a few steps—has therefore attracted growing attention (Boffi et al., 2025; Sabour et al., 2025). A variety of approaches have been explored, including improved numerical solvers (Tong et al., 2025; Zhang & Chen, 2023; Zhao et al., 2023), distillation-based acceleration (Yin et al., 2024; Sauer et al., 2024), and consistency-based training (Song & Dhariwal, 2024; Geng et al., 2025c; Lu & Song, 2025), all aimed at reducing the number of sampling steps. Despite this progress, aggressively reducing steps often amplifies errors along the sampling trajectory, resulting in artifacts, semantic inconsistency, and degraded realism (Karras et al., 2022; 2024). Achieving both high fidelity and fast sampling, thus, remains challenging.

Flow Map Matching (FMM) offers a promising path toward fast generation (Geng et al., 2025a; Zhang et al., 2025; Luo et al., 2025; Kim et al., 2025; Geng et al., 2025b). Instead of learning an instantaneous velocity field and approximating the probability-flow ODE with many small steps, FMM directly learns finite-time transport maps between arbitrary time intervals (Wang et al., 2025a; Frans et al., 2025). This enables large transitions along the generative trajectory, making one-step or few-step sampling feasible in principle (Hu et al., 2025). Recent methods, such as MeanFlow (Geng et al., 2025a), show that time-dependent transport maps can substantially reduce inference cost while preserving competitive quality. From this perspective, FMM offers an appealing abstraction: it predicts a finite-time map that moves noisy states toward cleaner ones, thereby bypassing iterative refinement.

Nevertheless, the practical generative capacity of FMMs is often underutilized, particularly in regimes with extremely small NFEs. Finite-time transport learning is inherently approximate: the learned flow maps incur *transport estimation errors*, leading to a mismatch between the model-induced trajectory and the ideal data-generating process (Zhang et al., 2025; Lee et al., 2025b; Geng et al., 2025b), closely related to the training–sampling discrepancy in diffusion models (Ning et al., 2023; Li & van der Schaar, 2024). While such errors can directly distort the endpoint prediction in one-step sampling, they can further accumulate and propagate across transitions in few-step settings. Moreover, standard FMM training is dominated by reconstruction-style regression approaches (e.g., average-velocity matching), which encourage local alignment but may fail to capture the semantic and perceptual structure required for realistic synthesis (Li et al., 2023; Yu et al., 2025; Xiang et al., 2023). As a result, even when regression losses are small, generated samples can still deviate from the data manifold in visually significant ways (Yang & Wang, 2023; Chen et al., 2025b).

In this work, we ask: *How can an FMM model be refined so that its objective better reflects the sampling behavior executed at inference time?* We put forward **Contrastive**

**Flow Map Matching (CFMM)**, a principled framework that aligns FMM training with the model-induced flow maps, as illustrated in Figure 1. Our approach is motivated by an analysis of the distributional gap induced by transport errors. Specifically, we show that the reverse KL divergence between the target data distribution and the FMM-induced distribution admits an upper bound consisting of two interpretable components: (i) a *marginal mismatch* due to errors in intermediate transported states, and (ii) a *conditional mismatch* due to the final transition mapping intermediate states to clean samples. This decomposition suggests that improving fast sampling requires jointly correcting transport dynamics and refining the endpoint reconstruction under imperfect intermediates.

Guided by this insight, CFMM optimizes two complementary objectives. First, we retain average-velocity matching as a principled surrogate to improve transport dynamics and reduce marginal mismatch. Second, we introduce a sampling-aligned contrastive objective that serves as a practical surrogate for endpoint-level conditional consistency. Rather than enforcing strict pixel-level reconstruction— often brittle under inevitable transport estimation errors— we compare model outputs along the sampling trajectory to their ground-truth targets using a feature-space similarity score optimized via InfoNCE (Oord et al., 2018). This provides robust supervision for semantic consistency (e.g., identity, object presence, layout) even when exact reconstruction is imperfect, while encouraging discriminative representations that complement regression-based learning.

In short, CFMM is simple and practical. It can be applied as a plug-in fine-tuning procedure for pre-trained FMMs with moderate additional training cost and *no inference-time overhead*. It can also be used to train FMMs from scratch, offering a general recipe for sampling-aligned transport learning. We validate CFMM on CIFAR-10, ImageNet, and LSUN across multiple FMM baselines, observing consistent improvements in sample fidelity and perceptual quality under one- and few-step sampling.

## 2. Preliminary

This section briefly reviews the theoretical foundations of flow matching (FM) and flow map models, which serve as the core building blocks of our framework.

**Flow Matching.** The goal of generative modeling is to transform a simple prior distribution (typically $\mathcal{N}(0, I)$) into the target data distribution $p_{\text{data}}(x)$ through a time-evolving transport process. FM instantiates this idea by learning a continuous-time velocity field that governs the evolution of samples along a trajectory from noise to data. Concretely, given a data sample $x \sim p_{\text{data}}(x)$ and a noise

sample $e \sim \mathcal{N}(0, I)$, FM considers a linear interpolation:

$$z_t = (1-t)x + te, \quad t \in [0, 1].$$

Differentiating $z_t$ yields the corresponding instantaneous velocity $v_t \triangleq \frac{dz_t}{dt} = e - x$. FM trains a neural network $v_\theta(z_t, t)$ by minimizing a regression loss in velocity space:

$$\mathcal{L}_{\text{FM}}(\theta) = \mathbb{E}_{t,x,e}\left[\left\|v_\theta(z_t, t) - (e - x)\right\|^2\right].$$

Since the same intermediate state $z_t$ can arise from multiple pairs $(x, e)$, the optimal predictor under squared loss corresponds to the marginal velocity $v_t(z_t, t) \triangleq \mathbb{E}[v_t \mid z_t]$, which averages over all compatible $(x, e)$ given $(z_t, t)$. After optimization, samples are generated by solving the probability-flow ODE $\frac{dz_t}{dt} = v_{\theta^*}(z_t, t)$ with initialization $z_1 \sim p_{\text{prior}}$ and integration from $t = 1$ to $t = 0$. In practice, accurate numerical integration typically requires many discretization steps, which limits sampling efficiency.

**Flow Map Models.** To reduce the reliance on multi-step ODE integration, flow map models have been proposed as an alternative paradigm for fast generation. Rather than predicting an instantaneous velocity, they aim to parameterize finite-time transport maps that directly connect states across arbitrary time intervals, enabling coarse or even one-step transitions along the generative trajectory. Here, we use MeanFlow (Geng et al., 2025a) as a representative example.

Let $v(z_t, t)$ denote the instantaneous velocity field. Mean-Flow defines the average velocity between time points $r$ and $t$ as:

$$u(z_t, r, t) \triangleq \frac{1}{t-r} \int_r^t v(z_\tau, \tau)\, d\tau.$$

Direct evaluation of this integral is generally intractable during training (Geng et al., 2025a;b). To obtain a tractable learning objective, MeanFlow differentiates the above definition with respect to $t$ and derives the identity:

$$u(z_t, r, t) = v(z_t, t) - (t - r)\frac{du(z_t, r, t)}{dt}. \quad (1)$$

In Eq. (1), $v(z_t, t)$ is instantiated as $e - x$ (as in FM), leaving $\frac{du(z_t, r, t)}{dt}$ as the only unknown term. This derivative can be computed efficiently via a Jacobian–vector product (JVP) between the Jacobian $[\partial_z u, \partial_r u, \partial_t u]$ and the tangent vector $[v, 0, 1]$. For brevity, we denote this operation by $\text{JVP}(u; v)$, evaluated at $u(z_t, r, t)$ and $v(z_t, t)$, yielding:

$$\text{JVP}(u; v) \triangleq \partial_z u(z_t, r, t)\, v(z_t, t) + \partial_t u(z_t, r, t).$$

MeanFlow then employs a neural network $u_\theta(z_t, r, t)$ to approximate the average-velocity field and optimizes it to satisfy Eq. (1). Concretely, the target is instantiated as:

$$u_{\text{tgt}} = (e - x) - (t - r)\,\text{JVP}(u_\theta; e - x), \quad (2)$$

where two approximations are adopted (Geng et al., 2025a): (i) the marginal velocity $v(z_t, t)$ is replaced by the conditional estimate $e - x$ (as in flow matching), and (ii) the ground-truth $u$ inside the JVP is replaced by the network prediction $u_\theta$. Given this target, MeanFlow minimizes the following regression objective, where $\text{sg}(\cdot)$ denotes the stop-gradient operator for stabilizing training:

$$\mathcal{L}_{\text{average}} = \mathbb{E}_{t,r,x,e}\left[\left\|u_\theta(z_t, r, t) - \text{sg}(u_{\text{tgt}})\right\|^2\right], \quad (3)$$

After training converges to $\theta^*$, MeanFlow enables a direct transition from $z_t$ to time $r$ via $\hat{z}_r = z_t - (t - r)\,u_{\theta^*}(z_t, r, t)$.

## 3. Theoretical Analysis

Although MeanFlow can yield high-quality one-step generation, its training is inherently limited by *velocity estimation error*, which can constrain its modeling capacity. This issue is not specific to MeanFlow and also appears in other FMM variants. In what follows, we first clarify the origin of this error and then derive a KL upper bound that yields a principled decomposition of the modeling gap, motivating tractable surrogate objectives.

### 3.1. Velocity Estimation Error

MeanFlow learns an average-velocity field $u(z_t, r, t)$ by regressing a network $u_\theta(z_t, r, t)$ to a training target $u_{\text{tgt}}$. A key ingredient in constructing $u_{\text{tgt}}$ is the Jacobian–vector product $\text{JVP}(u; v)$, which ideally depends on the ground-truth average velocity $u(z_t, r, t)$. In practice, however, MeanFlow replaces this unknown quantity with the network prediction $u_\theta(z_t, r, t)$ when computing $\text{JVP}(u; v)$. As a result, the training target becomes self-referential: it depends on the current model parameters.

This approximation is particularly problematic early in training, when $u_\theta(z_t, r, t)$ is inaccurate due to random initialization. Consequently, the constructed target can deviate from the ideal average velocity,

$$u_{\text{tgt}} \neq u(z_t, r, t) = \frac{1}{t-r} \int_r^t v(z_\tau, \tau)\, d\tau,$$

introducing an intrinsic bias into the regression objective. In principle, one would like to replace $u_\theta$ in $\text{JVP}(u_\theta; e - x)$ with the ground-truth $u(z_t, r, t)$. However, this is infeasible because the exact integral $\frac{1}{t-r} \int_r^t v(z_\tau, \tau)\, d\tau$ is generally unknown. Therefore, even after optimization, $u_{\theta^*}(z_t, r, t)$ may fail to faithfully recover the true average velocity, leading to inevitable velocity estimation error.

At inference time, this error directly impacts sampling: inaccuracies in $u_{\theta^*}$ distort the model-induced trajectory and consequently perturb the generated sample $\hat{x}$ away from the data manifold. This induces a distributional mismatch between the model-induced distribution $q_\theta(\hat{x})$ and

the target distribution $p_{\text{data}}(x)$, which can be quantified by $D_{\text{KL}}(p_{\text{data}}(x) \,\|\, q_\theta(\hat{x}))$. However, directly minimizing this divergence is intractable because $q_\theta(\hat{x})$ is only implicitly defined through the learned transport dynamics.

### 3.2. Theoretical Decomposition

Since the ground-truth counterpart of a generated sample is not available at inference time, minimizing sample-wise reconstruction discrepancies is generally infeasible. Instead, we consider the distributional gap between the target data distribution $p_{\text{data}}(x)$ and the model-induced distribution $q_\theta(\hat{x})$, measured by $D_{\text{KL}}(p_{\text{data}}(x) \,\|\, q_\theta(\hat{x}))$. Directly optimizing this divergence is challenging because $q_\theta(\hat{x})$ is implicitly defined by the learned flow-map sampling process and does not admit an explicit likelihood. We therefore first derive a joint-KL upper bound that decomposes the terminal modeling gap into two interpretable components. This decomposition serves as the theoretical motivation for the surrogate objectives introduced in the next section.

**Theorem 3.1.** *Let $z_r = (1-r)x + re$, where $x \sim p_{\text{data}}(x)$ and $e \sim \mathcal{N}(0, I)$, and let $P(x, z_r)$ denote the induced joint distribution over $(x, z_r)$. Similarly, let $Q_\theta(\hat{x}, \hat{z}_r)$ denote the joint distribution over $(\hat{x}, \hat{z}_r)$ induced by the flow-map model, where $\hat{z}_r$ is the model-generated intermediate state. Assume that $P$ and $Q_\theta$ are defined on the same endpoint and intermediate-state spaces, admit regular conditional distributions, and that $P$ is absolutely continuous with respect to $Q_\theta$. Then, the following upper bound holds:*

$$\begin{aligned} D_{\text{KL}}(p_{\text{data}}(x) \,\|\, q_\theta(\hat{x})) &\leq D_{\text{KL}}(P(x, z_r) \,\|\, Q_\theta(\hat{x}, \hat{z}_r)) \\ &= D_{\text{KL}}(p(z_r) \,\|\, q_\theta(\hat{z}_r)) \\ &+ \mathbb{E}_{z \sim p(z_r)}[D_{\text{KL}}(p(x \mid z_r = z) \,\|\, q_\theta(\hat{x} \mid \hat{z}_r = z))]. \end{aligned} \quad (4)$$

*Proof sketch.* The above inequality follows from the data-processing property of KL divergence by marginalizing the joint distributions onto their endpoint variables. Applying the chain rule of KL divergence to the joint distributions then yields the decomposition in Eq. (4). The first term, $D_{\text{KL}}(p(z_r) \,\|\, q_\theta(\hat{z}_r))$, measures the discrepancy between the ground-truth intermediate distribution and the model-induced intermediate distribution. The second term, $\mathbb{E}_{z \sim p(z_r)}[D_{\text{KL}}(p(x \mid z_r = z) \,\|\, q_\theta(\hat{x} \mid \hat{z}_r = z))]$, characterizes the endpoint conditional discrepancy after identifying $z_r$ and $\hat{z}_r$ as variables taking values in the same intermediate-state space. Here, $q_\theta(\hat{x} \mid \hat{z}_r = z)$ denotes the regular conditional distribution of the model-induced joint law evaluated at the same intermediate-state value $z$; this does not imply that $\hat{z}_r$ is sampled from $p(z_r)$, but follows from the KL chain rule with $p(z_r)$ as the reference marginal. A detailed proof is provided in the Appendix. $\square$

Theorem 3.1 provides an exact decomposition of a joint-KL upper bound on the terminal modeling gap. It reveals two complementary sources of training–sampling discrepancy in flow-map sampling: a marginal mismatch over intermediate states and a conditional mismatch in endpoint reconstruction. However, the two KL terms in Eq. (4) are generally not directly tractable, since the model-induced marginal and conditional distributions are only implicitly specified by the learned sampling dynamics. Therefore, we do not claim to directly minimize these KL terms. Instead, we use this decomposition as a guiding principle to design tractable surrogate objectives: average-velocity regression for improving intermediate transport consistency, and a sampling-aligned contrastive objective for encouraging endpoint-level semantic consistency.

## 4. Contrastive Flow Map

To unlock the underutilized generative capacity of FMMs, we propose a sampling-aligned learning framework that refines the model along its own induced flow-map trajectory. The central idea is to optimize tractable objectives that correspond to the two sources of mismatch identified in Theorem 3.1, rather than directly minimizing the intractable KL terms themselves. Specifically, we retain average-velocity regression as a surrogate for improving intermediate transport consistency, and introduce a sampling-aligned contrastive objective as a surrogate for improving endpoint-level semantic consistency under model-induced intermediate states. Guided by this decomposition, our framework improves fast-sampling performance and can be applied either as a plug-in refinement for pre-trained FMMs or as a training objective from scratch.

In what follows, we first characterize the distributional mismatch induced by imperfect finite-time transport maps. We then introduce tractable surrogate objectives motivated by the marginal and conditional components in the joint-KL decomposition, and finally present the resulting overall optimization objective.

### 4.1. Overview

We begin by formalizing the training–sampling mismatch of FMMs at the distributional level. Let $x \sim p_{\text{data}}$ and $e \sim p_{\text{prior}}$ be paired data and noise samples, and define the standard linear bridge: $z_t = (1-t)x + te, t \in (0, 1)$. For any $r < t$, an FMM parameterizes an average-velocity field $u_\theta(\cdot, \cdot, \cdot)$, which induces the finite-time transport operator:

$$\Phi_\theta^{r \leftarrow t}(z_t) \triangleq z_t - (t-r)u_\theta(z_t, r, t), \quad \hat{z}_r = \Phi_{\theta^*}^{r \leftarrow t}(z_t).$$

where $\theta^*$ denotes the optimized parameters. Under the ground-truth bridge, the latent state at time $r$ is given by $z_r \triangleq (1-r)x + re$. Ideally, the learned map $\Phi_{\theta^*}^{r \leftarrow t}$ should match the true transition $z_t \mapsto z_r$. However, approximation errors in $u_{\theta^*}$ generally lead to a discrepancy between the induced *pushforward* distributions. Specifically, letting $p_t$

denote the marginal distribution of $z_t$, the model-induced marginal at time $r$ becomes: $q_\theta(\hat{z}_r) \triangleq (\Phi_\theta^{r\leftarrow t})_\# p_t$, where $(\Phi_\theta)_\# p$ denotes the pushforward measure under $\Phi_\theta$.

Such mismatch propagates to the final generated sample. In particular, composing the learned transport to the endpoint yields $\hat{x} \triangleq \Phi_{\theta^*}^{0\leftarrow r}(\hat{z}_r)$, $q_\theta(\hat{x}) = (\Phi_\theta^{0\leftarrow r} \circ \Phi_\theta^{r\leftarrow t})_\# p_t$, which defines the model distribution $q_\theta(\hat{x})$ only implicitly through the sampling dynamics. Since $\hat{x}$ is not paired with a unique ground-truth $x$ at inference time, directly minimizing sample-wise reconstruction losses is not feasible. Instead, our goal is to reduce the distributional gap between $p_{\text{data}}(x)$ and $q_\theta(\hat{x})$, e.g., via $D_{\text{KL}}(p_{\text{data}}(x)\|q_\theta(\hat{x}))$. However, this objective is intractable to optimize directly because $q_\theta(\hat{x})$ does not admit an explicit likelihood.

As shown in Theorem 3.1, the terminal distributional gap can be upper-bounded by a joint KL that decomposes into two components: a marginal term, $D_{\text{KL}}(p(z_r)\|q_\theta(\hat{z}_r))$, and an endpoint conditional term, $\mathbb{E}_{z\sim p(z_r)}\left[D_{\text{KL}}(p(x \mid z_r = z) \| q_\theta(\hat{x} \mid \hat{z}_r = z))\right]$. This decomposition identifies two sources of training–sampling discrepancy in FMMs: imperfect intermediate transport and imperfect endpoint reconstruction from model-induced intermediate states. However, neither component is directly tractable to optimize in practice, because the model-induced marginal and conditional distributions are only implicitly defined by the learned sampling dynamics. We therefore use this decomposition as a guiding principle to design practical surrogate objectives for improving intermediate transport consistency and endpoint semantic consistency.

### 4.2. Surrogate Objectives

The decomposition in Theorem 3.1 suggests that improving FMM sampling requires addressing both intermediate transport mismatch and endpoint reconstruction mismatch. Since the corresponding KL terms are intractable for likelihood-free flow-map sampling, we introduce tractable surrogate objectives for these two components. For the marginal component, average-velocity regression encourages the model-induced intermediate state $\hat{z}_r$ to match the ground-truth bridge state $z_r$. For the conditional component, we introduce a sampling-aligned InfoNCE objective that encourages the endpoint generated from a model-induced trajectory to remain semantically consistent with its paired data sample in a frozen representation space. Importantly, the contrastive objective is used as a surrogate for endpoint-level conditional consistency, rather than as a direct minimization of the original conditional KL term.

#### 4.2.1. AVERAGE-VELOCITY OBJECTIVE

**Lemma 4.1.** *Assume that the model-induced transition distribution follows a Gaussian form, $q_\theta(\hat{z}_r \mid z_t) = \mathcal{N}(\hat{z}_r; \mu_\theta(z_t, r, t), \sigma_r^2 I)$, where $\sigma_r > 0$, $z_t = (1-t)x + te$,*

*and $\mu_\theta(z_t, r, t)$ the state-level transition mean induced by the FMM. Then, minimizing the conditional KL $D_{\text{KL}}(p(z_r \mid z_t)\|q_\theta(\hat{z}_r \mid z_t))$ is equivalent (up to a $\theta$-independent constant) to minimizing the following regression objective:*

$$\mathbb{E}_{p(z_t, z_r)} \left[\|z_r - \mu_\theta(z_t, r, t)\|^2\right]. \tag{5}$$

*Consequently, the same equivalence holds for the marginal objective $\mathbb{E}_{p(z_t)}\left[D_{\text{KL}}(p(z_r \mid z_t)\|q_\theta(\hat{z}_r \mid z_t))\right]$.*

*Proof.* By definition, we have $D_{\text{KL}}(p\|q_\theta) = \mathbb{E}_p[\log p] - \mathbb{E}_p[\log q_\theta]$, where the first term is independent of the model parameters $\theta$. Moreover, $-\log q_\theta(z_r \mid z_t) = \frac{1}{2\sigma_r^2}\|z_r - \mu_\theta(z_t, r, t)\|^2 + c$, where $c$ is a constant. Taking the expectation over $p(z_t, z_r)$ then yields Eq. (5). $\square$

**Proposition 4.2.** *Assume the ground-truth one-step transition from time $t$ to $r$ is given by $z_r = z_t - (t - r)u(z_t, r, t)$ and the model transition is parameterized as $\hat{z}_r = z_t - (t - r)u_\theta(z_t, r, t)$. Further assume that the model-induced transition distribution follows the Gaussian form $q_\theta(\hat{z}_r \mid z_t) = \mathcal{N}(\hat{z}_r; \mu_\theta(z_t, r, t), \sigma_r^2 I)$, where $\mu_\theta(z_t, r, t) = z_t - (t - r)u_\theta(z_t, r, t)$ and $\sigma_r > 0$. If the involved conditional KL divergences are finite, then there exists a constant $c$ independent of $\theta$ such that*

$$D_{\text{KL}}(p(z_r) \| q_\theta(\hat{z}_r))$$
$$\leq \mathbb{E}_{p(z_t)}\left[D_{\text{KL}}(p(z_r \mid z_t) \| q_\theta(\hat{z}_r \mid z_t))\right]$$
$$\leq \frac{(t-r)^2}{2\sigma_r^2}\mathbb{E}_{p(z_t, z_r)}\left[\|u_\theta(z_t, r, t) - u(z_t, r, t)\|^2\right] + c.$$

*Proof.* The result follows by first applying the data-processing property to marginalize out $z_t$, and then using Lemma 4.1 together with $\mu_\theta(z_t, r, t) = z_t - (t - r)u_\theta(z_t, r, t)$. Here, $\mu_\theta(z_t, r, t)$ denotes the state-level transition **mean** induced by the FMM, whereas $u_\theta(z_t, r, t)$ denotes the predicted average velocity. $\square$

In summary, Proposition 4.2 provides an assumption-based justification for using average-velocity regression as a surrogate for marginal alignment. Under the Gaussian transition model, reducing the average-velocity estimation error decreases a transition-level upper bound on the intermediate marginal mismatch. This supports the use of $\mathcal{L}_{\text{average}}$ for improving the model-induced intermediate state $\hat{z}_r$. However, this objective mainly regularizes the intermediate transport dynamics and does not explicitly constrain the endpoint mapping from $\hat{z}_r$ to $\hat{x}$, leaving endpoint-level conditional consistency only indirectly addressed.

#### 4.2.2. CONTRASTIVE OBJECTIVE

While $\mathcal{L}_{\text{average}}$ improves intermediate transport consistency, it does not explicitly encourage the final sample generated

from a model-induced intermediate state to preserve the semantic content of the paired data sample. The conditional term in Theorem 3.1 motivates this issue, but directly optimizing it is infeasible because the model-induced conditional distribution $q_\theta(\hat{x} \mid \hat{z}_r = z)$ is implicitly defined by the sampling dynamics. We therefore introduce a sampling-aligned contrastive objective as a tractable surrogate for endpoint-level conditional consistency.

To clarify the role of this objective, we introduce an auxiliary energy-based conditional model in representation space. This auxiliary model is not assumed to be identical to the original model-induced conditional distribution in Theorem 3.1. Rather, it provides a tractable contrastive surrogate that encourages matched pairs $(\hat{x}, x)$ generated from the same training trajectory to have higher semantic similarity than mismatched pairs.

By definition, the conditional term can be reformulated as:

$$
\begin{aligned}
D_{\mathrm{KL}}(p(x \mid z_r) \| q_\theta(\hat{x} \mid \hat{z}_r)) = \\
\mathbb{E}_{p(x \mid z_r)}[\log p(x \mid z_r) - \log q_\theta(\hat{x} \mid \hat{z}_r)].
\end{aligned} \tag{6}
$$

Since $q_\theta(\hat{x} \mid \hat{z}_r)$ is implicitly defined by the sampling dynamics, directly optimizing Eq. (6) is generally intractable. To circumvent this issue, we introduce a learnable critic $T_\psi(\hat{x}, x)$ and define an auxiliary energy-based conditional distribution $q_\psi(x \mid \hat{x})$ as in Eq. (7). This construction turns conditional mismatch minimization into optimizing matched-pair scores while controlling the log-normalizer.

Given a critic $T_\psi(\hat{x}, x)$, we define an auxiliary energy-based conditional distribution:

$$
\begin{aligned}
q_\psi(x \mid \hat{x}) &\triangleq \frac{p_{\mathrm{data}}(x) \exp(T_\psi(\hat{x}, x))}{Z_\psi(\hat{x})}, \\
Z_\psi(\hat{x}) &\triangleq \mathbb{E}_{x \sim p_{\mathrm{data}}}\left[\exp(T_\psi(\hat{x}, x))\right].
\end{aligned} \tag{7}
$$

Intuitively, Eq. (7) parameterizes a family of conditional distributions over $x$ given $\hat{x}$, whose log density is:

$$
\log q_\psi(x \mid \hat{x}) = \log p_{\mathrm{data}}(x) + T_\psi(\hat{x}, x) - \log Z_\psi(\hat{x}). \tag{8}
$$

This auxiliary distribution should be interpreted as a representation-level proxy for endpoint consistency, rather than as an explicit model of the original conditional distribution $q_\theta(\hat{x} \mid \hat{z}_r)$.

**Lemma 4.3** (Auxiliary Conditional Decomposition)**.** *Under the definition in Eq. (7), the auxiliary endpoint-consistency discrepancy satisfies* $\mathbb{E}_{p(z_r)}[D_{\mathrm{KL}}(p(x \mid z_r) \| q_\psi(x \mid \hat{x}))] = \mathbb{E}_{p(z_r, x)}[-T_\psi(\hat{x}, x) + \log Z_\psi(\hat{x})] + C$, *where* $C = \mathbb{E}_{p(z_r)}[D_{\mathrm{KL}}(p(x \mid z_r) \| p_{\mathrm{data}}(x))]$ *is independent of* $\psi$. *Here, the expectation is taken over the joint training construction that produces the matched pair* $(\hat{x}, x)$.

*Proof sketch.* First, we expand $D_{\mathrm{KL}}(p(x \mid z_r) \| q_\psi(x \mid \hat{x}))$ by applying Eq. (8) together with the identity $D_{\mathrm{KL}}(p \| q) =$

$\mathbb{E}_p[\log p - \log q]$, and then take expectation over $p(z_r)$. Accordingly, all $\psi$-independent terms are absorbed into the constant $C$. More derivations are provided in the appendix. $\square$

Lemma 4.3 shows that reducing the conditional mismatch can be achieved by increasing the critic score $T_\psi(\hat{x}, x)$ for matched pairs while controlling the normalizer $\log Z_\psi(\hat{x})$. Since $\log Z_\psi(\hat{x})$ involves an expectation over $p_{\mathrm{data}}(x)$, we approximate it via negative sampling and adopt the InfoNCE (Oord et al., 2018) objective:

$$
\mathcal{L}_{\mathrm{NCE}} = -\mathbb{E}\left[\log \frac{\exp(T_\psi(\hat{x}, x))}{\exp(T_\psi(\hat{x}, x)) + \sum_{i=1}^K \exp(T_\psi(\hat{x}, x_i^-))}\right]
$$

where $(\hat{x}, x)$ denotes the matched (positive) image pair, and $\{x_i^-\}_{i=1}^K$ are i.i.d. negative samples drawn from $p_{\mathrm{data}}(x)$.

**Proposition 4.4** (Auxiliary InfoNCE Surrogate)**.** *Assume the critic satisfies* $\exp(T_\psi(\hat{x}, x)) \in [0, B]$ *for some* $B > 0$. *Let* $(\hat{x}, x)$ *be a matched (positive) pair and let* $\{x_i^-\}_{i=1}^K$ *be i.i.d. negatives drawn from* $p_{\mathrm{data}}(x)$. *Define the empirical normalizer using negative samples,* $\widehat{Z}_\psi(\hat{x}) \triangleq \frac{1}{K}\sum_{i=1}^K \exp(T_\psi(\hat{x}, x_i^-))$. *Then, with probability at least* $1 - \delta$ *over the sampling of* $\{x_i^-\}_{i=1}^K$, *we have* $Z_\psi(\hat{x}) \leq \widehat{Z}_\psi(\hat{x}) + \epsilon_{K,\delta}$ *and* $\epsilon_{K,\delta} \triangleq B\sqrt{\frac{\log(1/\delta)}{2K}}$, *and consequently:*

$$
\begin{aligned}
\mathbb{E}_{p(z_r)}[D_{\mathrm{KL}}(p(x \mid z_r) \| q_\psi(x \mid \hat{x}))] \leq \mathcal{L}_{\mathrm{NCE}} - \\
\log K + \Delta_{K,\delta} + C,
\end{aligned}
$$

*where* $C$ *is a constant and* $K$ *denotes the number of negative samples. Moreover, the slack term can be defined as* $\Delta_{K,\delta} \triangleq \mathbb{E}_{p(z_r, x)}\left[\log\left(1 + \frac{\epsilon_{K,\delta}}{\widehat{Z}_\psi(\hat{x})}\right)\right]$. $\Delta_{K,\delta} \to 0$ *as* $K \to \infty$ *provided that* $\widehat{Z}_\psi(\hat{x})$ *remains bounded away from* 0 *with high probability.*

*Proof sketch.* By Lemma 4.3, it suffices to upper bound $\log Z_\psi(\hat{x})$, where $Z_\psi(\hat{x}) = \mathbb{E}_{x' \sim p_{\mathrm{data}}}[\exp(T_\psi(\hat{x}, x'))]$. Since $\exp(T_\psi(\hat{x}, x')) \in [0, B]$, Hoeffding's inequality implies that, with probability at least $1 - \delta$, $Z_\psi(\hat{x}) \leq \widehat{Z}_\psi(\hat{x}) + \epsilon_{K,\delta}$, and thus $\log Z_\psi(\hat{x}) \leq \log \widehat{Z}_\psi(\hat{x}) + \log\left(1 + \frac{\epsilon_{K,\delta}}{\widehat{Z}_\psi(\hat{x})}\right)$. Substituting this bound into Lemma 4.3 and using $\mathcal{L}_{\mathrm{NCE}} = \mathbb{E}\left[-T_\psi(\hat{x}, x) + \log \widehat{Z}_\psi(\hat{x})\right] + \log K$, completes the proof. Details are deferred to the appendix. $\square$

Proposition 4.4 shows that, under the auxiliary energy-based conditional model, the InfoNCE objective controls an auxiliary endpoint-consistency discrepancy up to a finite-sample slack term. This result should not be interpreted as a direct upper bound on the original model-induced conditional KL term in Theorem 3.1. Instead, it provides a principled surrogate interpretation: minimizing $\mathcal{L}_{\mathrm{NCE}}$ encourages the generated endpoint $\hat{x}$ to be closer to its paired data sample $x$ than to negative samples in a frozen semantic feature

space. Thus, the contrastive objective complements average-velocity regression by explicitly regularizing endpoint-level semantic consistency along the model-induced sampling trajectory.

### 4.3. Overall Optimization

We now summarize the training objective implied by the above analysis. Proposition 4.2 shows that average-velocity regression provides a principled surrogate for reducing the marginal mismatch $D_{\mathrm{KL}}(p(z_r)\|q_\theta(\hat{z}_r))$. Proposition 4.4 further establishes that an InfoNCE-style contrastive objective upper bounds the conditional mismatch $\mathbb{E}_{p(z_r)}[D_{\mathrm{KL}}(p(x \mid z_r)\|q_\theta(\hat{x} \mid \hat{z}_r))]$, up to a finite-sample slack term. Motivated by these results, we optimize an FMM by jointly minimizing both objectives to reduce the overall discrepancy $D_{\mathrm{KL}}(p_{\mathrm{data}}(x)\|q_\theta(\hat{x}))$.

To instantiate $\mathcal{L}_{\mathrm{NCE}}$, we parameterize the critic as a temperature-scaled feature similarity:

$$T_\psi(\hat{x}, x) \triangleq s_\psi(\hat{x}, x) = \frac{\langle f_\psi(\hat{x}),\, f_\psi(x)\rangle}{\tau}, \qquad \tau > 0,$$

where $f_\psi(\cdot)$ is a feature encoder and $\tau$ is a temperature hyper-parameter. In practice, we instantiate $f_\psi(\cdot)$ using a pre-trained DINOv2 model (Oquab et al., 2024) and freeze $\psi$ during training. From Proposition 4.4, the conditional mismatch is upper-bounded by $(\mathcal{L}_{\mathrm{NCE}} - \log K + \Delta_{K,\delta} + C)$, where the negatives $\{x_i^-\}_{i=1}^K$ in $\mathcal{L}_{\mathrm{NCE}}$ are sampled from other examples within the same training batch, $\Delta_{K,\delta}$ denotes the finite-sample slack, and $C$ collects terms independent of the learnable parameters. The positive pair $(\hat{x}, x)$ is constructed by sampling $z_t$ along the linear bridge between $x$ and $e$ and generating $\hat{x} \triangleq \Phi_\theta^{0\leftarrow r}\big(\Phi_\theta^{r\leftarrow t}(z_t)\big)$. Since $(-\log K + \Delta_{K,\delta} + C)$ is constant with respect to $\theta$, minimizing the bound is equivalent to minimizing $\mathcal{L}_{\mathrm{NCE}}$. Formally, letting $\mathcal{U}(\theta) \triangleq \mathcal{L}_{\mathrm{NCE}}(\theta) - \log K + \Delta_{K,\delta} + C$, we have:

$$\nabla_\theta \mathcal{U}(\theta) = \nabla_\theta \mathcal{L}_{\mathrm{NCE}}(\theta), \quad \arg\min_\theta \mathcal{U} = \arg\min_\theta \mathcal{L}_{\mathrm{NCE}}.$$

Therefore, optimizing the bound induces identical parameter updates to directly minimizing $\mathcal{L}_{\mathrm{NCE}}$. Putting everything together, we obtain the final training objective:

$$\mathcal{L}_{\mathrm{total}} = \mathcal{L}_{\mathrm{average}} + \lambda \mathcal{L}_{\mathrm{NCE}},$$

where $\lambda > 0$ balances marginal alignment and conditional refinement. During optimization, we update $\theta$ by minimizing $\mathcal{L}_{\mathrm{total}}$ while freezing $\psi$. By jointly reducing both surrogate terms in the KL upper bound, the proposed objective explicitly aligns practical sampling with training and improves FMM generation under fast sampling.

## 5. Experiments

This section mainly reports experimental results on fine-tuning various FMM variants. We also train several base-

*Table 1.* **Training MeanFlow from Scratch on ImageNet** $256 \times 256$. To verify the flexibility of our framework, we also train two MeanFlow variants, MeanFlow-B/2 and MeanFlow-XL/2, from random initialization. Under the same batch size and number of training iterations, our framework achieves better performance, and can be further improved by allocating additional training budget.

| Models | FID↓ | NFE↓ | Iter. | Batch | #Params |
|---|---|---|---|---|---|
| MeanFlow-B/2 | 6.04 | 1 | 300k | 256 | 131M |
| MeanFlow-B/2 | 5.17 | 2 | 300k | 256 | 131M |
| +ours | 5.88 | 1 | 300k | 256 | 131M |
| +ours | 5.09 | 2 | 300k | 256 | 131M |
| +ours | 5.01 | 4 | 300k | 256 | 131M |
| +ours | 4.98 | 8 | 300k | 256 | 131M |
| +ours | 5.74 | 1 | 400k | 256 | 131M |
| +ours | 5.02 | 2 | 400k | 256 | 131M |
| +ours | 4.95 | 4 | 400k | 256 | 131M |
| +ours | 4.90 | 8 | 400k | 256 | 131M |
| MeanFlow-XL/2 | 3.47 | 1 | 300k | 256 | 675M |
| MeanFlow-XL/2 | 2.46 | 2 | 300k | 256 | 675M |
| +ours | 2.95 | 1 | 300k | 256 | 675M |
| +ours | 2.24 | 2 | 300k | 256 | 675M |
| +ours | 2.17 | 4 | 300k | 256 | 675M |
| +ours | 2.10 | 8 | 300k | 256 | 675M |
| +ours | 2.90 | 1 | 400k | 256 | 675M |
| +ours | 2.06 | 2 | 400k | 256 | 675M |
| +ours | 2.03 | 4 | 400k | 256 | 675M |
| +ours | 2.00 | 8 | 400k | 256 | 675M |

*Table 2.* **Training MeanFlow from Scratch on Unconditional CIFAR-10.** Compared with SoTA baselines initialized from pre-trained EDM (Karras et al., 2022), including iCT (Song & Dhariwal, 2024), ECT (Geng et al., 2025c), sCT (Lu & Song, 2025), and IMM (Zhou et al., 2025), our framework achieves remarkable performance while outperforming the original MeanFlow.

| Models | precond | NFE↓ | FID↓ |
|---|---|---|---|
| iCT | EDM | 1 | 2.83 |
| ECT | EDM | 1 | 3.60 |
| sCT | EDM | 1 | 2.97 |
| IMM | EDM | 1 | 3.20 |
| MeanFlow | none | 1 | 2.92 |
| MeanFlow+ours | none | 1 | 2.70 |

lines from scratch to systematically evaluate the effectiveness of our framework. Below, we first describe the implementation details, and then report improvements over strong baselines, followed by comprehensive ablation studies.

**Implementation Details.** For fine-tuning, we strictly follow the original training configurations of all pre-trained baselines, only modifying the contrastive component as well as the batch size and training iterations. For training from scratch, we retain the default hyper-parameters of each baseline and incorporate the proposed contrastive objective with different $\lambda$ values. We instantiate the contrastive loss using a pre-trained DINOv2 encoder (Oquab et al., 2024; Yu

*Table 3.* **System-level Fine-tuning Evaluation on Class-conditional ImageNet** $256 \times 256$**.** To verify the effectiveness of our framework, we conduct comprehensive experiments on several advanced baselines, including Shortcut (Frans et al., 2025), MeanFlow (Geng et al., 2025a), TiM (Wang et al., 2025a), and $\alpha$-Flow (Zhang et al., 2025). Notably, the pre-trained MeanFlow parameters are provided by $\alpha$-Flow (Zhang et al., 2025). Clearly, our framework consistently improves performance while incurring only minimal extra training cost.

| Models | FID↓ | NFE↓ | Iter. | Batch | #Params |
|---|---|---|---|---|---|
| Shortcut-XL/2 | 10.60 | 1 | 800k | 256 | 675M |
| Shortcut-XL/2 | 7.8 | 4 | 800k | 256 | 675M |
| +ours | 9.74 | 1 | +40k | 256 | 675M |
| +ours | 8.65 | 2 | +40k | 256 | 675M |
| +ours | 7.01 | 4 | +40k | 256 | 675M |
| +ours | 6.39 | 8 | +40k | 256 | 675M |
| MeanFlow-B/2 | 6.04 | 1 | 300k | 256 | 131M |
| MeanFlow-B/2 | 5.17 | 2 | 300k | 256 | 131M |
| +ours | 5.76 | 1 | +35k | 256 | 131M |
| +ours | 5.05 | 2 | +35k | 256 | 131M |
| +ours | 5.03 | 4 | +35k | 256 | 131M |
| +ours | 5.01 | 8 | +35k | 256 | 131M |
| +ours | 5.71 | 1 | +40k | 512 | 131M |
| +ours | 5.00 | 2 | +40k | 512 | 131M |
| +ours | 5.71 | 1 | +40k | 1024 | 131M |
| +ours | 5.02 | 2 | +40k | 1024 | 131M |
| MeanFlow-XL/2 | 3.47 | 1 | 300k | 256 | 675M |
| MeanFlow-XL/2 | 2.46 | 2 | 300k | 256 | 675M |
| +ours | 2.98 | 1 | +30k | 256 | 675M |
| +ours | 2.10 | 2 | +30k | 256 | 675M |
| +ours | 2.05 | 4 | +30k | 256 | 675M |
| +ours | 2.02 | 8 | +30k | 256 | 675M |

| Models | FID↓ | NFE↓ | Iter. | Batch | #Params |
|---|---|---|---|---|---|
| TiM-XL/2 | 7.11 | 1 | 750k | 512 | 664M |
| TiM-XL/2 | 6.14 | 2 | 750k | 512 | 664M |
| TiM-XL/2 | 3.61 | 4 | 750k | 512 | 664M |
| TiM-XL/2 | 2.62 | 8 | 750k | 512 | 664M |
| +ours | 6.87 | 1 | +35k | 512 | 664M |
| +ours | 5.93 | 2 | +35k | 512 | 664M |
| +ours | 3.42 | 4 | +35k | 512 | 664M |
| +ours | 2.39 | 8 | +35k | 512 | 664M |
| $\alpha$-Flow-XL/2 | 2.95 | 1 | 1200k | 256 | 676M |
| $\alpha$-Flow-XL/2 | 2.16 | 2 | 1200k | 256 | 676M |
| +ours | 2.81 | 1 | +50k | 256 | 676M |
| +ours | 2.09 | 2 | +50k | 256 | 676M |
| +ours | 2.06 | 4 | +50k | 256 | 676M |
| +ours | 2.05 | 8 | +50k | 256 | 676M |
| $\alpha$-Flow-XL/2+ | 2.58 | 1 | 1275k | 1024 | 676M |
| $\alpha$-Flow-XL/2+ | 1.95 | 2 | 1275k | 1024 | 676M |
| +ours | 2.48 | 1 | +50k | 1024 | 676M |
| +ours | 1.90 | 2 | +50k | 1024 | 676M |
| +ours | 1.89 | 4 | +50k | 1024 | 676M |
| +ours | 1.88 | 8 | +50k | 1024 | 676M |

*Table 4.* **Fine-tuning Evaluation on LSUN datasets.** To evaluate performance on consistency models (CT) (Song et al., 2023), we enhance the pre-trained models using the same batch size and fine-tune for an additional 50k and 60k iterations on LSUN Bedroom and LSUN Cat, respectively. Improvements across three metrics systematically demonstrate the superiority of our framework.

| METHOD | NFE↓ | FID↓ | Prec.↑ | Rec.↑ |
|---|---|---|---|---|
| **LSUN Bedroom** $256 \times 256$ | | | | |
| CT | 1 | 16.0 | 0.60 | 0.17 |
| CT | 2 | 7.85 | 0.68 | 0.33 |
| CT+ours | 1 | 14.21 | 0.63 | 0.20 |
| CT+ours | 2 | 7.13 | 0.70 | 0.33 |
| **LSUN Cat** $256 \times 256$ | | | | |
| CT | 1 | 20.7 | 0.56 | 0.23 |
| CT | 2 | 11.7 | 0.63 | 0.36 |
| CT+ours | 1 | 18.67 | 0.58 | 0.26 |
| CT+ours | 2 | 10.39 | 0.65 | 0.37 |

et al., 2025). We conduct experiments on CIFAR-10 and ImageNet $256 \times 256$ with FMM models, and on LSUN Bedroom/Cat $256 \times 256$ with consistency models, which can be interpreted as a variant of FMM. Inference is performed using the original samplers of each baseline, resulting in no

additional sampling overhead. More details are provided in Table 5, and qualitative samples are shown in Figure 4.

**Performance Evaluation.** We evaluate our framework in two regimes. *(i) Plug-in refinement:* we fine-tune strong FMM baselines to improve fast sampling with modest additional compute. Tables 3 and 4 show consistent gains across diverse baselines with only 30k–50k extra iterations and no changes to the sampling procedure. On class-conditional ImageNet $256 \times 256$ (Table 3), we improve FID for several competitive FMM variants (e.g., Shortcut, MeanFlow, TiM, and $\alpha$-Flow). On LSUN Bedroom/Cat (Table 4), fine-tuning consistency models improves FID as well as Precision and Recall. *(ii) From-scratch training:* we train FMM backbones from random initialization to validate the objective as a standalone paradigm, and also consider consistency models as a special case of FMM that maps an intermediate state directly to the endpoint. On CIFAR-10 (Table 2), MeanFlow achieves better one-step generation than the original objective, and similar improvements hold on ImageNet $256 \times 256$ (Table 1) across model sizes and NFEs, with gains increasing under longer training and multi-step sampling.

**Ablation Studies.** We conduct comprehensive ablation studies to validate the design choices of our framework using MeanFlow-B/2 provided by $\alpha$-Flow (Zhang et al., 2025). Specifically, we ablate the weight $\lambda$, the stop-gradient set-

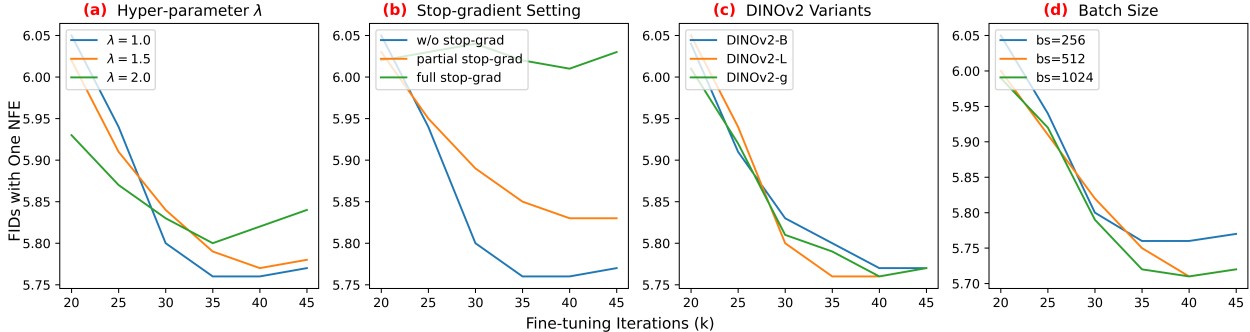

*Figure 2.* **Ablation Studies on Various Fine-tuning Techniques.** To conduct ablations, we use MeanFlow-B/2 as the backbone, with a batch size of 256 for all settings except (d). (a) We vary the weight $\lambda$ and find that $\lambda = 1.0$ performs best; we therefore use it in all experiments. (b) Our default setting allows gradients to propagate through both $0 \to r$ and $r \to t$ (*w/o stop-grad*). We additionally test *partial stop-grad*, which blocks gradients on $r \to t$ only, and *full stop-grad*, which blocks gradients on both transitions. (c) We compare different DINOv2 variants as the feature encoder and observe similar performance across all choices; thus, we adopt DINOv2-L for the remaining experiments. (d) We further test larger batch sizes and find that batch sizes of 512 and 1024 yield comparable improvements.

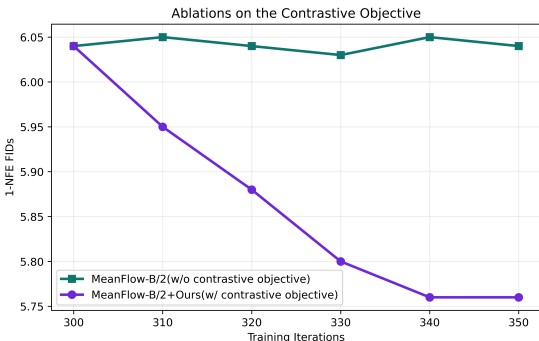

*Figure 3.* **Ablations on the Contrastive Objective.** Starting from the same pre-trained MeanFlow-B/2 checkpoint (300k iterations), we fine-tune MeanFlow-B/2 *without* the contrastive objective and with our contrastive objective, and report 1-NFE FIDs↓.

ting, different DINOv2 variants (DINOv2-B, DINOv2-L, and DINOv2-g), as well as the training iterations and batch size. As shown in Figure 2, our framework achieves the best performance with $\lambda = 1.0$ and without stop-gradient, and is relatively insensitive to the choice of DINOv2 variant. More ablation results are provided in Figure 3 and Table 6.

## 6. Conclusion

In this work, we propose an efficient training framework for FMMs that aligns optimization with model-induced flow maps to reduce the gap between training and inference. Motivated by a joint-KL decomposition, we interpret this gap as arising from intermediate marginal mismatch and endpoint conditional inconsistency, which leads to two practical surrogate objectives: average-velocity regression for improving transport consistency and a sampling-aligned InfoNCE loss for encouraging representation-level endpoint consistency. Experiments on CIFAR-10, ImageNet, and LSUN across multiple FMM baselines and consistency models show con-

sistent gains under one- and few-step sampling, with modest training cost and no inference-time overhead.

**Limitations and Future Work.** Our analysis should be interpreted as a decomposition-guided surrogate framework rather than a direct optimization of the original likelihood-free KL objective. Although Theorem 3.1 decomposes the terminal modeling gap into marginal and conditional components, these terms remain intractable for flow-map sampling. Thus, average-velocity matching and InfoNCE are used as practical surrogates for intermediate transport consistency and representation-level endpoint consistency. Further isolating the role of contrastive discrimination from feature supervision and establishing tighter links to the model-induced conditional KL remain important directions for future work, which we plan to investigate in subsequent studies.

## Acknowledgements

This work was supported by the Institute of Information & communications Technology Planning & Evaluation (IITP) grant funded by the Korean government (MSIT) (No. RS-2024-00457882, AI Research Hub Project, No.RS-2025-25441838, Development of a human foundation model for human-centric universal artificial intelligence and training of personnel, RS-2020-II201361, Artificial Intelligence Graduate School Program (Yonsei University)). This work was also supported in part by the Australian Research Council under Projects DP240101848 and FT230100549.

## Impact Statement

This paper presents work aimed at advancing the field of Machine Learning. There are many potential societal consequences of our work, none of which we feel must be specifically highlighted here.

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

# A. Related Works

**Diffusion and Flow-based Models.** Recent advances in Diffusion models (Song & Ermon, 2019; Ho et al., 2020; Song et al., 2021; Karras et al., 2022) have established them as a dominant framework for generative modeling in vision domain. These models operate by progressively corrupting clean data with noise and learning to approximate the associated reverse process. The sampling process involves solving a stochastic differential equation (SDE) (Song et al., 2021; Karras et al., 2022), which is typically approximated by numerical solvers (Song et al., 2020; Lu et al., 2022; Zhang & Chen, 2023; Zhao et al., 2023; Tong et al., 2025) in practice. Flow matching (Lipman et al., 2023) further extends this framework by adopting flow-based parameterization, which models a velocity field that flows from gaussian noise to the target distribution. While both diffusion and flow-based models have demonstrated remarkable performance across diverse generative tasks, their iterative sampling procedure leads to slow inference and computational inefficiency, which remains a major bottleneck for practical applications.

**Fastforward Generative Models.** Various attempts have been made to reduce the number of sampling steps of the diffusion and flow-based models. One line of this work is Consistency Models (Song et al., 2023; Lu & Song, 2025; Geng et al., 2025c), which enforce self-consistency to map any point at any time to the same initial point of the sampling trajectory. Either distilled from pre-trained diffusion models or trained from scratch, consistency models facilitate single-step sampling while also supporting multi-step sampling for quality-efficiency trade-offs. More recently, Flow Map models (Sabour et al., 2025; Geng et al., 2025a; Zhang et al., 2025; Wang et al., 2025a) have emerged as a promising approach for generative modeling. These models generalize existing consistency and flow matching objectives by enabling direct transitions between arbitrary time steps. A notable property of flow map models, in contrast to diffusion or flow-based models, is that they do not introduce discretization error during sampling, thereby achieving concrete fidelity improvements both on few- and many-step regimes.

**Representation Learning.** With the rapid progress of diffusion and flow-based models, recent works have investigated enhancing the representations learned during diffusion training. The pioneering work REPA (Yu et al., 2025) exploits a powerful pre-trained visual encoder (Oquab et al., 2024) to regularize diffusion models toward learning better representations, resulting in significant performance boost and faster training convergence. Subsequent works (Chen et al., 2025a; Leng et al., 2025; Lee et al., 2025a; Wang et al., 2025b; Tian et al., 2025; Wu et al., 2025a) explored improved scheduling, architectural adaptations, and joint training of the representation space under the REPA framework. Without relying on external encoders, (Wang & He, 2025) introduced Dispersive loss to encourage the dispersion of internal representations in the feature space, and demonstrated that contrastive-loss-like objective can benefit generative modeling.

# B. Theoretical Proofs.

In this section, we investigate how classifier-free guidance (CFG) integrates into our framework and present the corresponding theoretical derivations. Moreover, we also provide a detailed theoretical analysis of the main results to further improve the rigor and clarity of the paper.

## B.1. Incorporating Classifier-free Guidance.

We extend our framework to class-conditional generation using CFG training mechanism. Given a labeled sample $(x, c)$ and noise $e \sim p_{\text{prior}}$, we construct $z_t = (1 - t)x + te$. At inference time, CFG forms a guided average-velocity field by combining the conditional and unconditional predictions:

$$u_\theta^{\text{cfg}}(z_t, r, t \mid c; \gamma) \triangleq (1 + \gamma)\, u_\theta(z_t, r, t \mid c) - \gamma\, u_\theta(z_t, r, t \mid \varnothing), \qquad \gamma \geq 0,$$

which induces the guided one-step transport map $\Phi_{\theta,\text{cfg}}^{r \leftarrow t}(z_t \mid c) = z_t - (t - r)\, u_\theta^{\text{cfg}}(z_t, r, t \mid c; \gamma)$. For the value of $\gamma$, we follow the default setting used in the baselines to ensure a fair comparison. This is because our framework is designed to further improve modeling performance via the contrastive objective, rather than to tune or validate the choice of $\gamma$.

To align conditional sampling with ground-truth targets, we construct the sampling-aligned positive pair using the CFG-induced trajectory. Specifically, given $(x, c)$ and $e \sim p_{\text{prior}}$, we first sample $t$ and $r < t$, compute $z_t = (1 - t)x + te$, and apply the guided transport maps:

$$\hat{z}_r = \Phi_{\theta,\text{cfg}}^{r \leftarrow t}(z_t \mid c), \qquad \hat{x} = \Phi_{\theta,\text{cfg}}^{0 \leftarrow r}(\hat{z}_r \mid c).$$

We then form a positive pair $(\hat{x}, x)$ and optimize an InfoNCE-style objective using negatives $\{x_i^-\}_{i=1}^K$ sampled from other

data points:

$$\mathcal{L}_{\mathrm{NCE}}^{\mathrm{cfg}}(\theta) \triangleq -\mathbb{E}\left[\log \frac{\exp\big(T(\hat{x}, x)\big)}{\exp\big(T(\hat{x}, x)\big) + \sum_{i=1}^{K} \exp\big(T(\hat{x}, x_i^-)\big)}\right],$$

where $T(\hat{x}, x)$ is a feature-space critic (e.g., cosine similarity under DINO features). Finally, our conditional fine-tuning objective is

$$\min_{\theta} \ \mathcal{L}_{\mathrm{total}}^{\mathrm{cfg}}(\theta) \triangleq \mathcal{L}_{\mathrm{average}}^{\mathrm{cfg}}(\theta) + \lambda \, \mathcal{L}_{\mathrm{NCE}}^{\mathrm{cfg}}(\theta).$$

### B.2. Proof of Theorem 3.1.

*Proof.* We provide a detailed derivation of the claimed upper bound and KL decomposition.

**Notation.** Let $x \sim p_{\mathrm{data}}(x)$ and $e \sim \mathcal{N}(0, I)$ be independent, and define

$$z_r = (1-r)x + re.$$

This construction induces a joint distribution $p(x, z_r)$ with marginal $p(z_r)$ and conditional $p(x \mid z_r)$. Similarly, the flow-map model induces a joint distribution over the clean output and intermediate state, denoted by $q_\theta(\hat{x}, \hat{z}_r)$, with marginal $q_\theta(\hat{x})$ and $q_\theta(\hat{z}_r)$, and conditional $q_\theta(\hat{x} \mid \hat{z}_r)$.

**Step 1: Prove the upper bound.**

$$D_{\mathrm{KL}}(p_{\mathrm{data}}(x) \,\|\, q_\theta(\hat{x})) \ \leq \ D_{\mathrm{KL}}(p(x, z_r) \,\|\, q_\theta(\hat{x}, \hat{z}_r)). \tag{9}$$

This is a standard property that KL divergence cannot increase under marginalization. We prove it explicitly using the chain rule of KL. First, apply the KL chain rule to the joint distributions by conditioning on $x$:

$$\begin{aligned}
D_{\mathrm{KL}}(p(x, z_r) \,\|\, q_\theta(\hat{x}, \hat{z}_r)) &= \mathbb{E}_{p(x, z_r)}\left[\log \frac{p(x, z_r)}{q_\theta(\hat{x}, \hat{z}_r)}\right] \\
&= \mathbb{E}_{p(x, z_r)}\left[\log \frac{p(x)\, p(z_r \mid x)}{q_\theta(\hat{x})\, q_\theta(\hat{z}_r \mid \hat{x})}\right] \\
&= \mathbb{E}_{p(x, z_r)}\left[\log \frac{p(x)}{q_\theta(\hat{x})}\right] + \mathbb{E}_{p(x, z_r)}\left[\log \frac{p(z_r \mid x)}{q_\theta(\hat{z}_r \mid \hat{x})}\right].
\end{aligned} \tag{10}$$

The first term depends only on $x$, so

$$\mathbb{E}_{p(x, z_r)}\left[\log \frac{p(x)}{q_\theta(\hat{x})}\right] = \mathbb{E}_{p(x)}\left[\log \frac{p(x)}{q_\theta(\hat{x})}\right] = D_{\mathrm{KL}}(p_{\mathrm{data}}(x) \,\|\, q_\theta(\hat{x})). \tag{11}$$

The second term is a conditional KL averaged over $p(x)$:

$$\begin{aligned}
\mathbb{E}_{p(x, z_r)}\left[\log \frac{p(z_r \mid x)}{q_\theta(\hat{z}_r \mid \hat{x})}\right] &= \mathbb{E}_{p(x)}\left[\mathbb{E}_{p(z_r \mid x)}\left[\log \frac{p(z_r \mid x)}{q_\theta(\hat{z}_r \mid \hat{x})}\right]\right] \\
&= \mathbb{E}_{p(x)}\big[D_{\mathrm{KL}}\big(p(z_r \mid x) \,\|\, q_\theta(\hat{z}_r \mid \hat{x})\big)\big] \ \geq \ 0.
\end{aligned} \tag{12}$$

Combining (10)–(12) gives

$$D_{\mathrm{KL}}(p(x, z_r) \,\|\, q_\theta(\hat{x}, \hat{z}_r)) = D_{\mathrm{KL}}(p_{\mathrm{data}}(x) \,\|\, q_\theta(\hat{x})) + \mathbb{E}_{p(x)}\big[D_{\mathrm{KL}}\big(p(z_r \mid x) \,\|\, q_\theta(\hat{z}_r \mid \hat{x})\big)\big], \tag{13}$$

and since the second term is nonnegative, we obtain the desired upper bound (9).

**Step 2: Decompose the joint KL by the intermediate variable $z_r$.** We now derive the decomposition

$$D_{\mathrm{KL}}(p(x, z_r) \,\|\, q_\theta(\hat{x}, \hat{z}_r)) = D_{\mathrm{KL}}(p(z_r) \,\|\, q_\theta(\hat{z}_r)) + \mathbb{E}_{p(z_r)}\big[D_{\mathrm{KL}}\big(p(x \mid z_r) \,\|\, q_\theta(\hat{x} \mid \hat{z}_r)\big)\big]. \tag{14}$$

Starting from the definition,

$$
\begin{aligned}
D_{\mathrm{KL}}(p(x, z_r) \,\|\, q_\theta(\hat{x}, \hat{z}_r)) &= \mathbb{E}_{p(x, z_r)}\left[\log \frac{p(x, z_r)}{q_\theta(\hat{x}, \hat{z}_r)}\right] \\
&= \mathbb{E}_{p(x, z_r)}\left[\log \frac{p(z_r)\, p(x \mid z_r)}{q_\theta(\hat{z}_r)\, q_\theta(\hat{x} \mid \hat{z}_r)}\right] \\
&= \mathbb{E}_{p(x, z_r)}\left[\log \frac{p(z_r)}{q_\theta(\hat{z}_r)}\right] + \mathbb{E}_{p(x, z_r)}\left[\log \frac{p(x \mid z_r)}{q_\theta(\hat{x} \mid \hat{z}_r)}\right].
\end{aligned}
\tag{15}
$$

The first term depends only on $z_r$, hence

$$
\mathbb{E}_{p(x, z_r)}\left[\log \frac{p(z_r)}{q_\theta(\hat{z}_r)}\right] = \mathbb{E}_{p(z_r)}\left[\log \frac{p(z_r)}{q_\theta(\hat{z}_r)}\right] = D_{\mathrm{KL}}(p(z_r) \,\|\, q_\theta(\hat{z}_r)).
\tag{16}
$$

For the second term, take expectation over $p(x \mid z_r)$ inside:

$$
\begin{aligned}
\mathbb{E}_{p(x, z_r)}\left[\log \frac{p(x \mid z_r)}{q_\theta(\hat{x} \mid \hat{z}_r)}\right] &= \mathbb{E}_{p(z_r)}\left[\mathbb{E}_{p(x \mid z_r)}\left[\log \frac{p(x \mid z_r)}{q_\theta(\hat{x} \mid \hat{z}_r)}\right]\right] \\
&= \mathbb{E}_{p(z_r)}\left[D_{\mathrm{KL}}\big(p(x \mid z_r) \,\|\, q_\theta(\hat{x} \mid \hat{z}_r)\big)\right].
\end{aligned}
\tag{17}
$$

Substituting (16) and (17) into (15) yields (14).

**Conclusion.** Finally, combining Step 1 and Step 2, we obtain

$$
\begin{aligned}
D_{\mathrm{KL}}(p_{\mathrm{data}}(x) \,\|\, q_\theta(\hat{x})) &\leq D_{\mathrm{KL}}(p(x, z_r) \,\|\, q_\theta(\hat{x}, \hat{z}_r)) \\
&= D_{\mathrm{KL}}(p(z_r) \,\|\, q_\theta(\hat{z}_r)) + \mathbb{E}_{p(z_r)}\big[D_{\mathrm{KL}}\big(p(x \mid z_r) \,\|\, q_\theta(\hat{x} \mid \hat{z}_r)\big)\big],
\end{aligned}
\tag{18}
$$

which completes the proof. $\qquad\square$

### B.3. Proof of Lemma 4.1.

*Remark* B.1 (Connection to Gaussian MLE and extensions beyond isotropic variance). Lemma 4.1 is a direct consequence of the fact that, under a Gaussian likelihood with fixed variance, maximizing the conditional log-likelihood is equivalent to minimizing a squared error.

**Gaussian maximum likelihood view.** Fix $z_t$ and consider the conditional model $q_\theta(z_r \mid z_t) = \mathcal{N}(z_r; \mu_\theta(z_t, r, t), \sigma_r^2 I)$. Up to an additive constant independent of $\theta$, the negative log-likelihood is

$$
-\log q_\theta(z_r \mid z_t) = \frac{1}{2\sigma_r^2}\|z_r - \mu_\theta(z_t, r, t)\|^2 + c,
\tag{19}
$$

so minimizing $-\mathbb{E}_{p(z_r \mid z_t)}[\log q_\theta(z_r \mid z_t)]$ (equivalently, minimizing the conditional KL) reduces to least squares regression of $z_r$ onto $\mu_\theta(z_t, r, t)$.

**Diagonal covariance (anisotropic but factorized).** Suppose instead the model takes a diagonal Gaussian form

$$
q_\theta(z_r \mid z_t) = \mathcal{N}\Big(z_r; \mu_\theta(z_t, r, t), \mathrm{diag}(\sigma_\theta^2(z_t, r, t))\Big),
\tag{20}
$$

where $\sigma_\theta(z_t, r, t) \in \mathbb{R}_{>0}^d$ may be predicted by the model. Then the negative log-likelihood becomes

$$
-\log q_\theta(z_r \mid z_t) = \frac{1}{2}\sum_{i=1}^{d}\left(\frac{(z_{r,i} - \mu_{\theta,i}(z_t, r, t))^2}{\sigma_{\theta,i}^2(z_t, r, t)} + \log \sigma_{\theta,i}^2(z_t, r, t)\right) + c',
\tag{21}
$$

where $c' = \frac{d}{2}\log(2\pi)$ is constant. Consequently, minimizing the KL objective is equivalent to minimizing a *variance-weighted* regression loss with an additional $\log \sigma_{\theta,i}^2$ regularization term:

$$
\mathbb{E}_{p(z_t, z_r)}\left[\sum_{i=1}^{d}\left(\frac{(z_{r,i} - \mu_{\theta,i}(z_t, r, t))^2}{\sigma_{\theta,i}^2(z_t, r, t)} + \log \sigma_{\theta,i}^2(z_t, r, t)\right)\right].
\tag{22}
$$

In particular, if $\sigma_{\theta,i}$ is treated as fixed (not learned), the objective reduces to a weighted MSE.

**Full covariance (general Gaussian).** More generally, consider a Gaussian with full covariance

$$q_\theta(z_r \mid z_t) = \mathcal{N}(z_r; \mu_\theta(z_t, r, t), \Sigma_\theta(z_t, r, t)), \qquad \Sigma_\theta \succ 0. \tag{23}$$

Its negative log-likelihood is

$$-\log q_\theta(z_r \mid z_t) = \frac{1}{2}(z_r - \mu_\theta)^\top \Sigma_\theta^{-1}(z_r - \mu_\theta) + \frac{1}{2}\log \det \Sigma_\theta + \frac{d}{2}\log(2\pi). \tag{24}$$

Thus, the KL minimization objective becomes

$$\mathbb{E}_{p(z_t, z_r)}\left[(z_r - \mu_\theta(z_t, r, t))^\top \Sigma_\theta^{-1}(z_r - \mu_\theta(z_t, r, t)) + \log \det \Sigma_\theta(z_t, r, t)\right] \quad (+\text{const}), \tag{25}$$

which corresponds to minimizing a *Mahalanobis regression* term plus a log-determinant penalty.

**Interpretation.** The isotropic assumption $\Sigma = \sigma_r^2 I$ yields the simplest form, where KL minimization is exactly least squares (up to positive scaling and $\theta$-independent constants). Allowing non-isotropic or learnable covariance makes the objective heteroscedastic and introduces additional terms that control the predicted uncertainty.

### B.4. Proof of Proposition 4.2.

*Proof.* We provide a detailed derivation showing that the intermediate marginal mismatch $D_{\mathrm{KL}}(p(z_r) \,\|\, q_\theta(\hat{z}_r))$ can be upper bounded by a transition-level discrepancy, which further reduces to the mean squared average-velocity error under the Gaussian transition model.

**Step 1: From marginal KL to transition-level conditional KL.** Let

$$p(z_t, z_r) = p(z_t)p(z_r \mid z_t) \tag{26}$$

denote the ground-truth joint transition distribution induced by the bridge construction. We define the model-induced joint transition distribution as

$$q_\theta(z_t, \hat{z}_r) \triangleq p(z_t)q_\theta(\hat{z}_r \mid z_t), \tag{27}$$

where the two joint distributions share the same marginal $p(z_t)$ and differ only in the transition from $z_t$ to the intermediate state.

By the data-processing property of KL divergence, marginalizing out $z_t$ cannot increase the KL divergence. Therefore,

$$D_{\mathrm{KL}}(p(z_r) \,\|\, q_\theta(\hat{z}_r)) \leq D_{\mathrm{KL}}(p(z_t, z_r) \,\|\, q_\theta(z_t, \hat{z}_r)). \tag{28}$$

Using the factorization of the two joint distributions, we have

$$\begin{aligned}
&D_{\mathrm{KL}}(p(z_t, z_r) \,\|\, q_\theta(z_t, \hat{z}_r)) \\
&= \mathbb{E}_{p(z_t, z_r)}\left[\log \frac{p(z_t, z_r)}{q_\theta(z_t, \hat{z}_r)}\right] \\
&= \mathbb{E}_{p(z_t, z_r)}\left[\log \frac{p(z_t)p(z_r \mid z_t)}{p(z_t)q_\theta(z_r \mid z_t)}\right] \\
&= \mathbb{E}_{p(z_t, z_r)}\left[\log \frac{p(z_r \mid z_t)}{q_\theta(z_r \mid z_t)}\right] \\
&= \mathbb{E}_{p(z_t)}\left[D_{\mathrm{KL}}(p(z_r \mid z_t) \,\|\, q_\theta(\hat{z}_r \mid z_t))\right].
\end{aligned} \tag{29}$$

Here, $q_\theta(z_r \mid z_t)$ denotes the density of the model-induced transition distribution evaluated at the same intermediate-state value $z_r$; equivalently, it is the density associated with the random variable $\hat{z}_r$ conditioned on $z_t$.

Combining Eq. (28) and Eq. (29) gives

$$D_{\mathrm{KL}}(p(z_r) \,\|\, q_\theta(\hat{z}_r)) \leq \mathbb{E}_{p(z_t)}\left[D_{\mathrm{KL}}(p(z_r \mid z_t) \,\|\, q_\theta(\hat{z}_r \mid z_t))\right]. \tag{30}$$

**Step 2: Convert the conditional KL into a squared transition error.** Under the Gaussian transition assumption,

$$q_\theta(\hat{z}_r \mid z_t) = \mathcal{N}\left(\hat{z}_r; \mu_\theta(z_t, r, t), \sigma_r^2 I\right), \qquad \sigma_r > 0. \tag{31}$$

For fixed $z_t$, expanding the conditional KL gives

$$\begin{aligned}
&D_{\mathrm{KL}}(p(z_r \mid z_t) \,\|\, q_\theta(\hat{z}_r \mid z_t)) \\
&= \mathbb{E}_{p(z_r \mid z_t)} \left[\log p(z_r \mid z_t) - \log q_\theta(z_r \mid z_t)\right].
\end{aligned} \tag{32}$$

Since

$$-\log q_\theta(z_r \mid z_t) = \frac{1}{2\sigma_r^2} \|z_r - \mu_\theta(z_t, r, t)\|^2 + c_1, \tag{33}$$

where $c_1$ is independent of $\theta$, we obtain

$$\begin{aligned}
&D_{\mathrm{KL}}(p(z_r \mid z_t) \,\|\, q_\theta(\hat{z}_r \mid z_t)) \\
&= \frac{1}{2\sigma_r^2} \mathbb{E}_{p(z_r \mid z_t)} \left[\|z_r - \mu_\theta(z_t, r, t)\|^2\right] + c(z_t),
\end{aligned} \tag{34}$$

where $c(z_t)$ collects terms independent of $\theta$. Taking expectation over $p(z_t)$ yields

$$\begin{aligned}
&\mathbb{E}_{p(z_t)} \left[D_{\mathrm{KL}}(p(z_r \mid z_t) \,\|\, q_\theta(\hat{z}_r \mid z_t))\right] \\
&= \frac{1}{2\sigma_r^2} \mathbb{E}_{p(z_t, z_r)} \left[\|z_r - \mu_\theta(z_t, r, t)\|^2\right] + c_0,
\end{aligned} \tag{35}$$

where $c_0$ is independent of $\theta$.

**Step 3: Substitute the average-velocity parameterization.** By assumption, the ground-truth transition from $t$ to $r$ is

$$z_r = z_t - (t - r)u(z_t, r, t), \tag{36}$$

and the model-induced transition mean is

$$\mu_\theta(z_t, r, t) = z_t - (t - r)u_\theta(z_t, r, t). \tag{37}$$

Here, $\mu_\theta(z_t, r, t)$ denotes the state-level transition mean induced by the FMM, whereas $u_\theta(z_t, r, t)$ denotes the predicted average velocity. Therefore,

$$\begin{aligned}
z_r - \mu_\theta(z_t, r, t) &= \left(z_t - (t - r)u(z_t, r, t)\right) - \left(z_t - (t - r)u_\theta(z_t, r, t)\right) \\
&= (t - r)\left(u_\theta(z_t, r, t) - u(z_t, r, t)\right).
\end{aligned} \tag{38}$$

Consequently,

$$\|z_r - \mu_\theta(z_t, r, t)\|^2 = (t - r)^2 \|u_\theta(z_t, r, t) - u(z_t, r, t)\|^2. \tag{39}$$

Substituting Eq. (39) into Eq. (35) gives

$$\begin{aligned}
&\mathbb{E}_{p(z_t)} \left[D_{\mathrm{KL}}(p(z_r \mid z_t) \,\|\, q_\theta(\hat{z}_r \mid z_t))\right] \\
&= \frac{(t - r)^2}{2\sigma_r^2} \mathbb{E}_{p(z_t, z_r)} \left[\|u_\theta(z_t, r, t) - u(z_t, r, t)\|^2\right] + c_0.
\end{aligned} \tag{40}$$

**Step 4: Conclude the bound.** Combining Eq. (30) and Eq. (40), we obtain

$$\begin{aligned}
&D_{\mathrm{KL}}(p(z_r) \,\|\, q_\theta(\hat{z}_r)) \\
&\leq \mathbb{E}_{p(z_t)} \left[D_{\mathrm{KL}}(p(z_r \mid z_t) \,\|\, q_\theta(\hat{z}_r \mid z_t))\right] \\
&= \frac{(t - r)^2}{2\sigma_r^2} \mathbb{E}_{p(z_t, z_r)} \left[\|u_\theta(z_t, r, t) - u(z_t, r, t)\|^2\right] + c_0.
\end{aligned} \tag{41}$$

Setting $c \triangleq c_0$ completes the proof. $\qquad\square$

### B.5. Proof of Lemma 4.3.

*Proof.* We provide a detailed derivation of the stated decomposition.

**Recall the definition of $q_\psi(x \mid \hat{x})$.** By Eq. (7), the conditional distribution is defined via an energy-based (softmax) form:

$$q_\psi(x \mid \hat{x}) = \frac{p_{\text{data}}(x) \exp\left(T_\psi(\hat{x}, x)\right)}{Z_\psi(\hat{x})}, \qquad Z_\psi(\hat{x}) \triangleq \mathbb{E}_{p_{\text{data}}(x)}\left[\exp\left(T_\psi(\hat{x}, x)\right)\right]. \tag{42}$$

Equivalently, taking the logarithm yields Eq. (8):

$$\log q_\psi(x \mid \hat{x}) = \log p_{\text{data}}(x) + T_\psi(\hat{x}, x) - \log Z_\psi(\hat{x}). \tag{43}$$

**Step 1: Expand the conditional KL divergence pointwise in $z_r$.** For a fixed $z_r$, by the definition of KL divergence,

$$D_{\text{KL}}\left(p(x \mid z_r) \,\|\, q_\psi(x \mid \hat{x})\right) = \mathbb{E}_{p(x|z_r)}\left[\log \frac{p(x \mid z_r)}{q_\psi(x \mid \hat{x})}\right]$$
$$= \mathbb{E}_{p(x|z_r)}[\log p(x \mid z_r)] - \mathbb{E}_{p(x|z_r)}[\log q_\psi(x \mid \hat{x})]. \tag{44}$$

**Step 2: Substitute the log-form of $q_\psi$ and regroup terms.** Substitute (43) into (44):

$$D_{\text{KL}}\left(p(x \mid z_r) \,\|\, q_\psi(x \mid \hat{x})\right) = \mathbb{E}_{p(x|z_r)}[\log p(x \mid z_r)] - \mathbb{E}_{p(x|z_r)}\left[\log p_{\text{data}}(x) + T_\psi(\hat{x}, x) - \log Z_\psi(\hat{x})\right]$$
$$= \mathbb{E}_{p(x|z_r)}[\log p(x \mid z_r) - \log p_{\text{data}}(x)] - \mathbb{E}_{p(x|z_r)}[T_\psi(\hat{x}, x)] + \mathbb{E}_{p(x|z_r)}[\log Z_\psi(\hat{x})]. \tag{45}$$

Now observe that the first expectation in (45) is exactly a KL divergence:

$$\mathbb{E}_{p(x|z_r)}[\log p(x \mid z_r) - \log p_{\text{data}}(x)] = D_{\text{KL}}\left(p(x \mid z_r) \,\|\, p_{\text{data}}(x)\right), \tag{46}$$

which is independent of $\psi$.

Therefore (45) becomes

$$D_{\text{KL}}\left(p(x \mid z_r) \,\|\, q_\psi(x \mid \hat{x})\right) = D_{\text{KL}}\left(p(x \mid z_r) \,\|\, p_{\text{data}}(x)\right) + \mathbb{E}_{p(x|z_r)}\left[-T_\psi(\hat{x}, x) + \log Z_\psi(\hat{x})\right]. \tag{47}$$

**Step 3: Take expectation over $p(z_r)$ and simplify into a joint expectation.** Taking expectation over $z_r \sim p(z_r)$ on both sides of (47) yields

$$\mathbb{E}_{p(z_r)}\left[D_{\text{KL}}\left(p(x \mid z_r) \,\|\, q_\psi(x \mid \hat{x})\right)\right] = \mathbb{E}_{p(z_r)}\left[D_{\text{KL}}\left(p(x \mid z_r) \,\|\, p_{\text{data}}(x)\right)\right]$$
$$+ \mathbb{E}_{p(z_r)}\left[\mathbb{E}_{p(x|z_r)}\left[-T_\psi(\hat{x}, x) + \log Z_\psi(\hat{x})\right]\right]. \tag{48}$$

The nested expectation in the second term is equivalent to a joint expectation under $p(z_r, x) = p(z_r)\, p(x \mid z_r)$:

$$\mathbb{E}_{p(z_r)}\left[\mathbb{E}_{p(x|z_r)}\left[-T_\psi(\hat{x}, x) + \log Z_\psi(\hat{x})\right]\right] = \mathbb{E}_{p(z_r, x)}\left[-T_\psi(\hat{x}, x) + \log Z_\psi(\hat{x})\right]. \tag{49}$$

Define the constant

$$C \triangleq \mathbb{E}_{p(z_r)}\left[D_{\text{KL}}\left(p(x \mid z_r) \,\|\, p_{\text{data}}(x)\right)\right], \tag{50}$$

which is independent of $\psi$ since it does not involve $T_\psi$ nor $Z_\psi$.

Combining (48)–(50) gives

$$\mathbb{E}_{p(z_r)}\left[D_{\text{KL}}\left(p(x \mid z_r) \,\|\, q_\psi(x \mid \hat{x})\right)\right] = \mathbb{E}_{p(z_r, x)}\left[-T_\psi(\hat{x}, x) + \log Z_\psi(\hat{x})\right] + C, \tag{51}$$

which matches the statement of the lemma. $\square$

## B.6. Proof of Proposition 4.4.

*Proof.* We start from Lemma 4.3, which rewrites the conditional mismatch as

$$\mathbb{E}_{p(z_r)}[D_{\mathrm{KL}}(p(x \mid z_r)\|q_\psi(x \mid \hat{x}))] = \mathbb{E}_{p(z_r,x)}[-T_\psi(\hat{x}, x) + \log Z_\psi(\hat{x})] + C, \tag{52}$$

where

$$Z_\psi(\hat{x}) \triangleq \mathbb{E}_{x' \sim p_{\mathrm{data}}(x)}\big[\exp\big(T_\psi(\hat{x}, x')\big)\big], \tag{53}$$

and the constant

$$C \triangleq \mathbb{E}_{p(z_r)}[D_{\mathrm{KL}}(p(x \mid z_r)\|p_{\mathrm{data}}(x))] \tag{54}$$

is independent of $\psi$. Therefore, to upper bound the left-hand side, it suffices to upper bound the log-normalizer term $\log Z_\psi(\hat{x})$.

**Monte-Carlo estimator of the normalizer.** Let $\{x_i^-\}_{i=1}^K$ be i.i.d. negatives sampled from $p_{\mathrm{data}}(x)$. We define the empirical normalizer

$$\widehat{Z}_\psi(\hat{x}) \triangleq \frac{1}{K}\sum_{i=1}^K \exp\big(T_\psi(\hat{x}, x_i^-)\big). \tag{55}$$

For a fixed $\hat{x}$, introduce random variables

$$Y_i(\hat{x}) \triangleq \exp\big(T_\psi(\hat{x}, x_i^-)\big), \qquad i = 1, \dots, K. \tag{56}$$

Then $\{Y_i(\hat{x})\}_{i=1}^K$ are i.i.d. and satisfy

$$\mathbb{E}\big[Y_i(\hat{x}) \mid \hat{x}\big] = Z_\psi(\hat{x}), \qquad \frac{1}{K}\sum_{i=1}^K Y_i(\hat{x}) = \widehat{Z}_\psi(\hat{x}). \tag{57}$$

**A high-probability upper bound on $Z_\psi(\hat{x})$.** By assumption, the critic output satisfies $\exp(T_\psi(\hat{x}, x)) \in [0, B]$ for some $B > 0$. Hence for each fixed $\hat{x}$ we have $Y_i(\hat{x}) \in [0, B]$ almost surely. Hoeffding's inequality applied to the bounded i.i.d. variables $\{Y_i(\hat{x})\}$ yields that for any $\epsilon > 0$,

$$\mathbb{P}\Big(Z_\psi(\hat{x}) - \widehat{Z}_\psi(\hat{x}) \geq \epsilon \,\Big|\, \hat{x}\Big) \leq \exp\Big(-\frac{2K\epsilon^2}{B^2}\Big). \tag{58}$$

Setting the right-hand side to $\delta$ gives

$$\epsilon_{K,\delta} \triangleq B\sqrt{\frac{\log(1/\delta)}{2K}}, \tag{59}$$

and therefore, with probability at least $1 - \delta$ over the random draw of $\{x_i^-\}_{i=1}^K$ (conditioned on $\hat{x}$),

$$Z_\psi(\hat{x}) \leq \widehat{Z}_\psi(\hat{x}) + \epsilon_{K,\delta}. \tag{60}$$

**From an upper bound on $Z_\psi(\hat{x})$ to an upper bound on $\log Z_\psi(\hat{x})$.** On the event (60), and whenever $\widehat{Z}_\psi(\hat{x}) > 0$, we can take logarithms:

$$\log Z_\psi(\hat{x}) \leq \log\Big(\widehat{Z}_\psi(\hat{x}) + \epsilon_{K,\delta}\Big)$$

$$= \log \widehat{Z}_\psi(\hat{x}) + \log\Big(1 + \frac{\epsilon_{K,\delta}}{\widehat{Z}_\psi(\hat{x})}\Big). \tag{61}$$

The second equality uses the elementary identity $\log(a + b) = \log a + \log(1 + b/a)$ for $a > 0$.

**Substitution into the conditional KL decomposition.** Substituting (61) into (52) yields that, with probability at least $1 - \delta$,

$$\mathbb{E}_{p(z_r)}[D_{\mathrm{KL}}(p(x \mid z_r)\|q_\psi(x \mid \hat{x}))] \leq \mathbb{E}_{p(z_r,x)}\Big[-T_\psi(\hat{x}, x) + \log \widehat{Z}_\psi(\hat{x})\Big] + \Delta_{K,\delta} + C, \tag{62}$$

where we define the slack term

$$\Delta_{K,\delta} \triangleq \mathbb{E}_{p(z_r,x)}\left[\log\left(1 + \frac{\epsilon_{K,\delta}}{\widehat{Z}_\psi(\hat{x})}\right)\right]. \tag{63}$$

At this point, the only remaining task is to identify the first expectation term in (62) with an InfoNCE-style objective.

**Recovering the InfoNCE form and the appearance of** $-\log K$**.** By definition of $\widehat{Z}_\psi(\hat{x})$ in (55),

$$\log\widehat{Z}_\psi(\hat{x}) = \log\left(\frac{1}{K}\sum_{i=1}^{K}\exp(T_\psi(\hat{x}, x_i^-))\right)$$

$$= \log\left(\sum_{i=1}^{K}\exp(T_\psi(\hat{x}, x_i^-))\right) - \log K. \tag{64}$$

This identity is exactly where the $-\log K$ term originates: it is purely due to writing the log of an *average* as the log of a *sum* plus an additive $-\log K$.

Using (64), we define the InfoNCE-style loss (population form)

$$\mathcal{L}_{\text{NCE}} \triangleq \mathbb{E}_{p(z_r,x)}\left[-T_\psi(\hat{x},x) + \log\left(\sum_{i=1}^{K}\exp(T_\psi(\hat{x},x_i^-))\right)\right], \tag{65}$$

which implies the exact relation

$$\mathbb{E}_{p(z_r,x)}\left[-T_\psi(\hat{x},x) + \log\widehat{Z}_\psi(\hat{x})\right] = \mathcal{L}_{\text{NCE}} - \log K. \tag{66}$$

Substituting (66) into (62) establishes the desired bound:

$$\mathbb{E}_{p(z_r)}[D_{\text{KL}}(p(x \mid z_r)\|q_\psi(x \mid \hat{x}))] \leq \mathcal{L}_{\text{NCE}} - \log K + \Delta_{K,\delta} + C. \tag{67}$$

**Behavior of the slack term as** $K \to \infty$**.** From (59), we have $\epsilon_{K,\delta} = O(K^{-1/2}) \to 0$ as $K \to \infty$. If $\widehat{Z}_\psi(\hat{x})$ is bounded away from 0 with high probability, i.e., there exists $\eta > 0$ such that $\widehat{Z}_\psi(\hat{x}) \geq \eta$ holds with high probability, then

$$0 \leq \log\left(1 + \frac{\epsilon_{K,\delta}}{\widehat{Z}_\psi(\hat{x})}\right) \leq \log\left(1 + \frac{\epsilon_{K,\delta}}{\eta}\right) \to 0, \tag{68}$$

and hence $\Delta_{K,\delta} \to 0$ by dominated convergence (or by the squeeze argument above). This completes the proof. $\square$

## C. More Experiment Results.

In this section, we provide additional experimental results, including the main training configurations, further ablation studies, and qualitative visualizations.

**Training Configurations.** We summarize the training configurations of DiT-B/2, DiT-XL/2, and DiT-XL/2+ on ImageNet (256×256), including both model architectures and optimization hyperparameters (Table 5). We use DiT-B/2 as the default backbone for ablations and main-text analysis, and vary the number of training steps and batch size to study their effects on fine-tuning behavior. Across all settings, we adopt a unified optimization setup with Adam, a constant learning rate of $(1 \times 10^{-4})$, zero dropout and weight decay, EMA half-life 6931, gradient clipping norm 16, and the same autoencoder (sd-vae-ft-ema).

**More Ablation Studies.** Beyond the ablations on training techniques reported in the main experiments, we further investigate how the choice of loss metric affects the sampling-aligned objective. In our framework, each model prediction $\hat{x}$ is paired with a corresponding ground-truth target $x$, which allows the objective to be instantiated not only by the proposed contrastive loss, but also by a wide range of alternative regression-based metrics. To this end, we replace the InfoNCE objective with

*Table 5.* **Configurations on ImageNet** $256 \times 256$. DiT-B/2 serves as the backbone for our ablation and analysis in the main text. Moreover, we vary the training steps and batch size across different experiments.

| Configurations | DiT-B/2 | DiT-XL/2 | DiT-XL/2+ |
|---|---|---|---|
| *Network Architectures* | | | |
| Params (M) | 131 | 676 | 676 |
| Original FLOPs (G) | 23.1 | 119.0 | 119.0 |
| Fine-tuning FLOPs (G) | 34.2 | 141.5 | 141.5 |
| Depth | 12 | 28 | 28 |
| Hidden dim | 768 | 1152 | 1152 |
| Heads | 12 | 16 | 16 |
| Patch size | $2 \times 2$ | $2 \times 2$ | $2 \times 2$ |
| *Training hyperparameters* | | | |
| Pre-trained epochs (batch size) | 240 (256) | 240 (256) | 240(256)+60(1024) |
| Fine-tuning steps (batch size) | 35(256), 40(512), 40(1024) | 30(256), 50(256) | 50(1024) |
| Training from scratch steps (batch size) | 300(256), 400(256) | 300(256), 400(256) | – |
| Dropout | 0.0 | 0.0 | 0.0 |
| Optimizer | | Adam | |
| lr schedule | | constant | |
| lr | | 0.0001 | |
| Adam $(\beta_1, \beta_2)$ | | (0.9, 0.95) | |
| Weight decay | | 0.0 | |
| EMA half-life | | 6931 | |
| Gradient clipping norm | | 16 | |
| Autoencoder used | | sd-vae-ft-ema | |

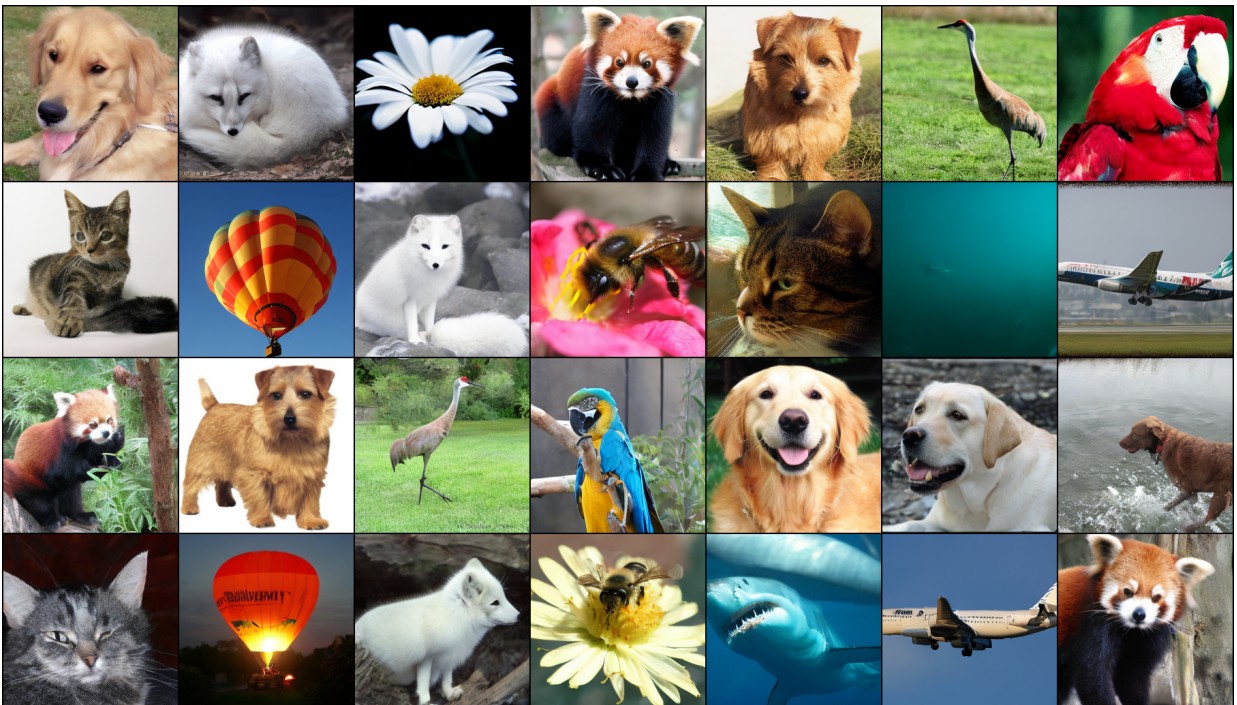

*Figure 4.* **ImageNet-**$256 \times 256$ **Samples Generated by MeanFlow-B/2 and Improved by Our Framework.**

several commonly used distance measures, including the $L_1$ norm, $L_2$ norm, LPIPS, and the Pseudo-Huber loss, while keeping all other training configurations unchanged. Additionally, we also report stronger baselines trained with more iterations and larger batch sizes but *without* the sampling-aligned objective, in order to ensure that the observed improvement

*Table 6.* **Ablations on Fine-Tuning MeanFlow-B/2 on ImageNet** $256 \times 256$ **with Different Metric Functions.** To comprehensively analyze the performance of using contrastive loss, we employ various loss functions to replace it and maintain other training settings as default. As shown above, our framework is effectiveness due to using contrastive objective.

| Models | NFEs=1 | NFEs=2 | Iteration | Batch Size | #Params |
|---|---|---|---|---|---|
| MeanFlow-B/2 | 6.04 | 5.17 | 300k | 256 | 131M |
| w/o extra loss | 6.05 | 5.18 | +35k | 256 | 131M |
| w/o extra loss | 6.02 | 5.16 | +40k | 512 | 131M |
| w/o extra loss | 6.01 | 5.15 | +40k | 1024 | 131M |
| +ours(w/ $L_1$ norm) | 6.53 | 6.05 | +35k | 256 | 131M |
| +ours(w/ $L_1$ norm) | 6.59 | 6.16 | +40k | 512 | 131M |
| +ours(w/ $L_1$ norm) | 6.64 | 6.29 | +40k | 1024 | 131M |
| +ours(w/ $L_2$ norm) | 7.11 | 6.15 | +35k | 256 | 131M |
| +ours(w/ $L_2$ norm) | 7.34 | 6.37 | +40k | 512 | 131M |
| +ours(w/ $L_2$ norm) | 7.50 | 6.64 | +40k | 1024 | 131M |
| +ours(w/ LPIPS) | 6.38 | 5.35 | +35k | 256 | 131M |
| +ours(w/ LPIPS) | 6.37 | 5.33 | +40k | 512 | 131M |
| +ours(w/ LPIPS) | 6.37 | 5.37 | +40k | 1024 | 131M |
| +ours(w/ Pseudo-Huber) | 6.23 | 5.24 | +35k | 256 | 131M |
| +ours(w/ Pseudo-Huber) | 6.25 | 5.20 | +40k | 512 | 131M |
| +ours(w/ Pseudo-Huber) | 6.19 | 5.19 | +40k | 1024 | 131M |
| +ours(w/ InfoNCE) | 5.76 | 5.05 | +35k | 256 | 131M |
| +ours(w/ InfoNCE) | 5.71 | 5.00 | +40k | 512 | 131M |
| +ours(w/ InfoNCE) | 5.71 | 5.02 | +40k | 1024 | 131M |

does not simply come from longer fine-tuning.

As summarized in Table 6, simply extending training without the additional objective yields only marginal gains (e.g., $6.04/5.17 \rightarrow 6.01/5.15$ for NFEs $= 1/2$), suggesting that longer optimization alone cannot overcome the bottleneck. Replacing the contrastive objective with regression-style metrics leads to limited improvement or even degradation: $L_1/L_2$ consistently worsen performance, while LPIPS and Pseudo-Huber are more stable but still remain inferior to InfoNCE. In contrast, our InfoNCE objective achieves the best results across all settings, improving MeanFlow-B/2 to $5.76/5.05$ with only $+35k$ iterations and further reaching $5.71/5.00$ with larger batch sizes and additional training. Overall, these results confirm that the improvements primarily stem from the sampling-aligned contrastive objective, rather than increased training budget or the particular choice of regression metric.

We attribute this behavior to the fact that the pre-trained FMM may fail to denoise certain intermediate states $\hat{z}_r$ into semantically meaningful outputs $\hat{x}$, especially when $r$ is far from the endpoint 0. In such cases, the regression loss between $\hat{x}$ and $x$ can become excessively large and noisy, providing poor learning signals and potentially destabilizing training. In contrast, the contrastive objective offers more robust supervision in representation space, making it better suited for correcting sampling-induced errors.

**Qualitative Visualizations.** We provide additional qualitative results on ImageNet-$256 \times 256$ to further demonstrate the effectiveness of our framework. As shown in Figure 4, our method consistently improves perceptual quality and semantic fidelity with limited sampling budgets, yielding sharper structures and more coherent object details without modifying the inference procedure or increasing the number of sampling steps. These results complement our quantitative evaluations and highlight the practical benefits of our framework for low-NFE generation.

