# OpenReview forum: "Contrastive Flow Map Matching"
_ICML.cc/2026/Conference — ICML 2026 regular_

### Official Review · Reviewer_eqdn · 2026-03-08

**Soundness:** 3
**Presentation:** 3
**Significance:** 2
**Originality:** 2
**Overall Recommendation:** 3
**Confidence:** 5

**Summary:**

In this paper, the authors propose Contrastive Flow Map Matching, an improvement to existing flop map methods like MeanFlows. Specifically, the authors augment the loss with an InfoNCE style loss, where:
(1) Original loss: given $x_t$, predict the average velocity $u_{t,r}$;
(2) InfoNCE loss: given $x_t = tx + (1-t)\epsilon$, follow $u_{t,r}$ and $u_{r,0}$ to achieve a sample $\hat x_0$, and encourage $x$ and $\hat x_0$ to be close while encouraging $\hat x_0$ to be different from other $x$ within a batch. Here, the distance is computed via cosine similarity of Dino embeddings.
Empirically, the authors find the method to improve performance when finetuning existing flow map models.

**Compliance With Llm Reviewing Policy:**

Affirmed.

**Final Justification:**

The approach augments Mean Flowwith a feature loss (the numerator) and a dispersive loss (the denominator). Given that perceptual and feature losses are widely known to directly improve FID, the marginal gains demonstrated in the fine-tuning experiments lack sufficient novelty and seem incremental.

**Key Questions For Authors:**

1. I'm wondering how does the authors justify the use of $\hat x$ being reconstructed from $z_r$. As said in the weakness session, I feel this is largely disconnected to the math derivations.
2. Empirically, I'm curious about how much performance gain comes from the use of perceptual loss, rather than the claim of "contrastive". Namely:
(1) If the loss is only defined on $|\hat x - x\|$ (in the space of Dino), how much gain is there? This is the perceptual loss baseline.
(2) If the loss is defined as the contrastive form, but doesn't use the Dino features, how much gain is there? And how does it compare to other contrastive regularization losses, say the dispersive loss in the paper "Diffuse and disperse: Image generation with representation regularization"?
3. How much additional compute does the InfoNCE loss incur? Seems the model has to back-propagate through the feature extractor, which sounds expensive .

**Limitations:**

yes

**Strengths And Weaknesses:**

Strengths:
1. The improvement is consistent across datasets and methods, showing a consistent trend.
2. It's encouraging to see the method works under finetuning settings.

Weaknesses:
1. The use of InfoNCE loss is not well justified. It's unclear that the loss aligns with the objective of equation (5). Namely, $\hat z_r$ is constructed by following $u_{t,r}$ from $x_t$, so the conditional distribution $p(\hat x | \hat z_r)$ should be the distribution of $x$ where $z_r=x_t - (t-r) u_{t,r}$, rather than the distribution of $z_r - r u_{0,r}$. It's unclear why the InfoNCE loss is applied on the reconstruction from $z_r$: such $x$ is only a single point, not even a distribution. This is very disconnected to the motivation from equation (5).
2. As a result, I highly suspect the InfoNCE loss is essentially doing a Dino-v2 perceptual loss, since the loss term asks the reconstruction to be close to data sample in Dino space. It's not surprising that perceptual losses improve generative modeling.
3. Also: the improvement in FID seems somewhat incremental.

---

> ### Author Rebuttal · Authors · 2026-03-31
>
> We thank the reviewer for the careful reading of our manuscript.
> ### W1&Q1. It's unclear that the loss aligns with the objective of equation (5).
> Our method indeed employs an InfoNCE objective, and the intended connection is the following. Eq. (5) motivates the problem at the distribution level by identifying the endpoint conditional mismatch after an imperfect intermediate transition. Since this conditional term is not directly tractable to optimize, we implement a sample-level, sampling-aligned InfoNCE objective in practice: we generate $\hat{x}$ along the model path $z\_t\rightarrow\hat{z}\_r\rightarrow\hat{x}$ and compare $\hat{x}$ with its matched target $x$.
>
> This choice is intentional. At inference time, the model never receives the ground-truth $z\_r$; it must map its own model-induced intermediate $\hat{z}\_r$ to the final sample. Therefore, supervision should be imposed on the **model-generated endpoint** $\hat{x}$, not on an idealized reconstruction from $z\_r$. In this sense, Eq. (5) identifies the relevant endpoint error, while InfoNCE is the tractable objective we use to improve that endpoint mapping in practice. Although each loss term is defined on a sampled pair $(\hat{x},x)$, the overall objective is optimized in expectation over sampled $(x,e,t,r)$ and thus acts as a practical surrogate for improving the endpoint mapping under model-induced intermediate errors.
> ### W2&Q2. How much performance gain comes from perceptual loss, rather than the claim of “contrastive”?
> We agree that it is important to disentangle the gain from **representation-based perceptual supervision** and that from the **contrastive formulation itself**.
> First, we keep the same pretrained feature space, namely **DINOv2-L**, and compare several objectives on MeanFlow-B/2 over ImageNet-256:
> |Method|NFE=1|NFE=2|
> |:-|:-:|:-:|
> |MeanFlow-B/2 (base)|6.04|5.17|
> |MeanFlow-B/2 (w/ Cos. sim.)|5.83|5.09|
> |MeanFlow-B/2 (w/ feature L2)|5.75|5.06|
> |MeanFlow-B/2 (w/ NT-Xent)|5.72|5.03|
> |MeanFlow-B/2 (w/ InfoNCE)|5.71|5.02|
>
> These results show that DINOv2-based representation supervision alone already yields clear gains over the base model, confirming that perceptual/semantic alignment is an important ingredient. At the same time, contrastive objectives further improve over plain representation matching, with InfoNCE achieving the best performance among these nearby alternatives.
>
> To further isolate the role of the contrastive formulation from that of pretrained DINOv2 features, we additionally consider a contrastive-without-DINO setting, where frozen DINOv2 features are replaced by internal MeanFlow features plus a lightweight projection head; all objectives below are computed in this same internal feature space:
> |Method|NFE=1|
> |:-|:-:|
> |MeanFlow-XL/2 (base)|3.43|
> |MeanFlow-XL/2 (w/ Disp) [1]|3.21|
> |MeanFlow-XL/2 (w/ Cos. sim.)|3.23|
> |MeanFlow-XL/2 (w/ feature L2)|3.15|
> |MeanFlow-XL/2 (w/ NT-Xent)|3.07|
> |MeanFlow-XL/2 (w/ InfoNCE)|3.05|
>
> This second ablation shows that contrastive learning still improves the baseline even without DINOv2, so the gain is not solely due to pretrained perceptual features. Overall, the evidence suggests that CFMM benefits from both ingredients: strong semantic supervision from pretrained representations and an additional improvement from the contrastive formulation, with the combination giving the best performance.
>
> [1] Diffuse and Disperse: Image Generation with Representation Regularization. arXiv 2025.
> ### W3. FID seems somewhat incremental.
> Our method is a training-only plug-in that requires only minor extra fine-tuning, introduces no inference-time overhead, and can also be used for training from scratch. In this context, the observed FID gains are already meaningful.
>
> Specifically, TiM-XL/2 uses only 35k extra iterations on top of 750k training iterations, and $\alpha$-Flow-XL/2 uses only 50k extra iterations on top of 1200k, i.e., less than 4% additional training in both cases. Even with such limited extra cost, CFMM consistently improves multiple strong FMM baselines. Moreover, these gains are not explained by extra optimization alone: simply extending training without the proposed objective yields only marginal improvement, whereas adding CFMM produces clear additional gains.
> ### Q3. How much additional compute does the InfoNCE loss incur?
> In practice, we freeze the DINOv2 parameters and update only the generators, so the computation does not increase excessively. To quantify the additional compute cost, we report the FLOPs in Table 5 of the appendix and summarize the performance-computation trade-off below:
> |Method|FID|FLOPs (G)|
> |:-|:-:|:-:|
> |MeanFlow-B/2|6.04|23.1|
> |MeanFlow-B/2+CFMM|5.71|34.2|
> |MeanFlow-XL/2|3.47|119.0|
> |MeanFlow-XL/2+CFMM|2.98|141.5|
>
> CFMM increases training-time compute moderately while consistently improving generation quality. Moreover, this extra cost is incurred only during training; the inference procedure remains unchanged and introduces no additional overhead.

---

> > ### Author Rebuttal · Reviewer_eqdn · 2026-04-04
> >
> > I thank the authors for the rebuttal and the response.
> > Based on the table above, most gains did come from perceptual loss, while the contrastive loss provides slightly additional gains (from 5.75 to 5.71 for B size, and from 3.15 to 3.05 for XL size). The later improvements seem somewhat incremental; and also, it can be attributed to the gap induced by dispersive loss. As a result, I will like to maintain my score.

---

> > > ### Author Response · Authors · 2026-04-06
> > >
> > > We sincerely thank the reviewer for the continued feedback.
> > >
> > > First, we would like to clarify a *potential misinterpretation* of the ablation tables, and we apologize for any confusion caused by the table formulation.
> > > Each column corresponds to training the base MeanFlow-XL/2 with **one additional loss only**, rather than cumulatively combining all losses.
> > > For example, **MeanFlow-XL/2 (w/ feature L2)** and **MeanFlow-XL/2 (w/ InfoNCE)** are two independent variants, corresponding to fine-tuning the pre-trained MeanFlow-XL/2 with **the MeanFlow objective + feature L2** and **the MeanFlow objective + InfoNCE**, respectively.
> > > Therefore, the improvement from the contrastive objective should be interpreted as **3.43 $\to$ 3.05**, rather than **3.15 $\to$ 3.05**, since **3.15** is already the result of using **feature L2** to fine-tune the MeanFlow-XL/2 baseline.
> > >
> > > Second, we agree that both perceptual loss and InfoNCE are defined in the same feature space.
> > > However, this does **not imply equivalence**.
> > > A perceptual loss enforces **absolute feature matching** between $\hat{x}$ and $x$, whereas InfoNCE introduces **relative discrimination**, encouraging $\hat{x}$ to be close to its matched target while remaining distinguishable from negative samples.
> > > This induces a discriminative structure in the feature space that feature regression alone does not provide.
> > >
> > > More importantly, the use of InfoNCE is **not an ad-hoc design choice**, but is directly motivated by our theoretical framework.
> > > Specifically, we show that the conditional mismatch term $\mathbb{E}\_{p(z\_r)}\left[D\_{\mathrm{KL}}(p(x\mid z\_r)\|q\_{\psi}(x\mid \hat{x}))\right]$ is intractable in practice, and InfoNCE provides a **tractable upper-bound surrogate** for this objective.
> > > Therefore, the contrastive loss is introduced to **target the conditional mismatch identified in the analysis**, rather than simply adding a perceptual regularizer.
> > >
> > > Finally, in our setting, the mapping from $z\_r$ to $x$ is inherently ambiguous: a single intermediate state may correspond to multiple plausible outputs with similar perceptual semantics.
> > > In such cases, enforcing pointwise perceptual similarity can be insufficient or even misleading.
> > > The contrastive objective instead encourages the model to **distinguish the correct target from competing alternatives**, which is more aligned with the conditional distribution we aim to approximate.
> > >
> > > This distinction becomes even clearer in the **training-from-scratch** setting, which is also an important use case of our method.
> > > At early stages of training, a non-converged model often fails to produce a semantically meaningful $\hat{x}$ from $z_r$.
> > > As a result, $\hat{x}$ and $x$ may not share reliable perceptual structure, making direct perceptual regression unstable or ineffective.
> > > To verify this, we additionally trained the baseline from scratch, i.e., with randomly initialized model parameters.
> > > The **FID** results are as follows:
> > > | Training Models from Scratch | NFE=1 |
> > > |:-|:-:|
> > > | MeanFlow-B/2 (base) | 6.04 |
> > > | MeanFlow-B/2 (MeanFlow loss + feature L2) | 20.75 |
> > > | MeanFlow-B/2 (MeanFlow loss + InfoNCE) | 5.74 |
> > > | MeanFlow-XL/2 (base) | 3.47 |
> > > | MeanFlow-XL/2 (MeanFlow loss + feature L2) | 15.92 |
> > > | MeanFlow-XL/2 (MeanFlow loss + InfoNCE) | 2.90 |
> > >
> > > These results show that, in this setting, simply applying a perceptual loss can severely harm optimization, whereas the proposed CFMM remains effective.
> > > This further suggests that our method should not be viewed as a generic perceptual regularizer, but rather as a theoretically motivated and practically robust objective for reducing the conditional mismatch in FMM training.
> > >
> > > We thank the reviewer again for the constructive feedback and sincerely hope this clarification will help the reviewer reassess the contribution of the paper.

---

### Official Review · Reviewer_nnvS · 2026-03-11

**Soundness:** 2
**Presentation:** 2
**Significance:** 2
**Originality:** 3
**Overall Recommendation:** 2
**Confidence:** 4

**Summary:**

This paper introduces CFMM, which adds a contrastive InfoNCE loss on top of average-velocity regression for training flow map models (FMMs). The motivation comes from decomposing the reverse KL between the data distribution and the model-induced distribution into a marginal mismatch over intermediate states and a conditional mismatch in endpoint mapping (Theorem 3.1). The marginal term is addressed by the existing MeanFlow regression objective; the conditional term is addressed by a new sampling-aligned InfoNCE loss computed in the feature space of a frozen DINOv2 encoder. The combined objective can fine-tune pre-trained FMMs or train them from scratch. Experiments on CIFAR-10, ImageNet 256x256, and LSUN show FID gains across four FMM baselines (Shortcut, MeanFlow, TiM, alpha-Flow) and consistency models, with 30k--50k extra training iterations and no change to the inference procedure.

**Compliance With Llm Reviewing Policy:**

Affirmed.

**Final Justification:**

I thank the authors for the additional ablations and comparisons, which are helpful. However, my main concerns remain. The theoretical contributions, i.e., originally presented as formal guarantees, have been downgraded across the board to "surrogates" and "proxies". For a paper that uses its theoretical framework as a central motivation, this retreat substantially weakens the contribution. I maintain my score.

**Key Questions For Authors:**

1. In Proposition 4.2's proof, how do you justify substituting $z\_r = z\_t - (t-r)u$ when the expectation in Lemma 4.1 is over the interpolation distribution where $z\_r = (1-r)x + re$? These differ for individual samples by exactly the velocity estimation error.

2. Proposition 4.4 bounds the KL with $q\_\psi$ (the energy-based surrogate), not the KL with $q\_\theta$ from Theorem 3.1. Can you formally connect these two quantities? Without this link, the theoretical chain from data KL to the training loss is incomplete.

3. Have you tested the contrastive loss with a randomly initialized encoder or with CLIP instead of DINOv2? This would clarify whether the gains stem from the contrastive mechanism itself or from DINOv2's representations.

4. How does the framework extend to multi-step sampling? The bound covers a single $z\_t \to z\_r \to x$ decomposition, but experiments go up to 8 NFEs. Does the bound degrade multiplicatively with the number of steps?

5. Can you provide a direct comparison with REPA applied to the same FMM baselines? Both methods use frozen DINOv2 for training-time alignment, so the comparison is natural and necessary.

**Strengths And Weaknesses:**

Strengths:

- The method is genuinely simple to use without architectural changes or inference overhead. This kind of plug-in recipe has real practical value.

- FID improvements are consistent across four distinct FMM baselines on ImageNet (Table 1), consistency models on LSUN (Table 4), and from-scratch MeanFlow training on CIFAR-10 and ImageNet (Tables 2,3).

- The ablations in Figure 3 and Table 6 are informative. Table 6 is particularly useful: it shows that extending training without the contrastive loss yields marginal FID gains (6.04 to 6.01 for MeanFlow-B/2 at 1 NFE), while adding InfoNCE reaches 5.76. It also shows InfoNCE consistently outperforms $L\_1$, $L\_2$, LPIPS, and Pseudo-Huber alternatives as the contrastive metric. These ablations are more convincing than the theory.

Weaknesses:

- The proof of Proposition 4.2 (page 5, lines 249--254; Appendix B.4 Step 3) contains an error. The proof substitutes $z\_r = z\_t - (t-r)u$ where $u$ is the ground-truth average velocity, treating $z\_r$ as deterministic given $z\_t$. But in Lemma 4.1, the MSE expectation is over the training interpolation distribution where $z\_r = (1-r)x + re$ and different $(x,e)$ pairs sharing the same $z\_t$ produce different $z\_r$ values. For a single sample, $z\_r - \mu\_\theta$ equals $(t-r)$ times the difference $u\_\theta - (e-x)$, not $u\_\theta - u$. A bias-variance decomposition gives the correct expression: $(t-r)^2$ times the sum of $\mathbb{E}[\\|u\_\theta - v\_t\\|^2]$ and the irreducible conditional variance of $e - x$ given $z\_t$, where $v\_t$ is the marginal velocity at time $t$ -- not the average velocity $u$ used in the proposition. The variance term is $\theta$-independent and gets absorbed into the constant, but the bound should involve $v\_t$, not $u$.

- There is a disconnect in the theoretical chain between Theorem 3.1 and the actual training objective. Theorem 3.1 decomposes the KL gap into a conditional term involving $q\_\theta$, the model's implicit transition distribution (Eq. 5). Since this is intractable, the paper introduces an energy-based surrogate $q\_\psi$ (Eq. 6) and proceeds to bound a different KL involving $q\_\psi$ instead (Lemma 4.3, Proposition 4.4). These are different quantities: one conditions on $\hat{z}\_r$ through the model dynamics, the other conditions on $\hat{x}$ through a learned critic, and no formal relationship between them is established. The paper claims on lines 318--319 that the InfoNCE bounds the conditional mismatch from Theorem 3.1, but Proposition 4.4 actually bounds the surrogate KL with $q\_\psi$. The chain from $D\_\mathrm{KL}(p\_\mathrm{data} \\| q\_\theta)$ to the training loss is broken at this step.

- Even setting aside the proof issues, the bound framework appears vacuous. The marginal bound (Proposition 4.2) has a $1/\sigma\_r^2$ prefactor where $\sigma\_r$ is a fictitious Gaussian variance that does not exist in the actual deterministic FMM. As $\sigma\_r \to 0$ (the regime matching the real model), the bound diverges. The conditional bound (Proposition 4.4) retains $C$, defined as the expected KL between $p(x|z\_r)$ and $p\_\mathrm{data}(x)$ (Eq. 47), which is independent of all learnable parameters and could be arbitrarily large. No estimate of these constants is provided, and there is no evidence the bound is ever informative.

- The theory addresses only 2-step transitions: $z\_t \to z\_r \to x$. Experiments evaluate up to 8 NFEs. Extending the bound to multi-step sampling would require recursive decomposition at each intermediate step, and the looseness would compound. The paper does not discuss this. During training, random $(t, r)$ pairs are sampled, which implicitly covers different step sizes, but the theory does not justify why this should help at 4 or 8 NFEs.

- I cannot tell whether the improvements come from the contrastive learning framework or from distilling DINOv2 features into the generative model. REPA (Yu et al., 2025) showed that a plain regression loss aligning intermediate representations with DINOv2 already yields large gains in flow-based generation. The paper cites REPA in the related work (Appendix A, line 629) but never compares against it experimentally. The DINOv2 ablation in Figure 3(c) only swaps model sizes within the DINOv2 family. Testing a fundamentally different encoder (CLIP, MAE, or even a random network) would help isolate whether InfoNCE is doing the heavy lifting or DINOv2 is.

- The main paper also lacks fixed-seed qualitative comparisons. It should show the results from same noise input through the baseline and the proposed method side by side. Without this, FID improvements could reflect distributional shifts rather than per-sample quality gains.


[R1] Yu et al. "Representation alignment for generation: Training diffusion transformers is easier than you think." ICLR, 2025.

---

> ### Author Rebuttal · Authors · 2026-03-31
>
> ## W1. Proposition 4.2
> We thank the reviewer for catching this. We agree that the current proof of Proposition 4.2 is too strong as written and will revise it using a more careful conditional-mean / bias–variance argument.
>
> This correction does **not** change the overall CFMM framework. Theorem 3.1 still provides the core decomposition into **marginal mismatch** and **conditional mismatch**, which remains the main motivation of the method. The only change is interpretational: the practical MeanFlow-style regression should be viewed as a **surrogate** for reducing marginal mismatch, rather than as an exact bound in its current form.
>
> ## W2&Q2. Formal connection between $q\_\psi$ and $q\_\theta$
> We agree that the current draft omits an explicit linking step. Proposition 4.4 upper-bounds a surrogate conditional KL involving $q_\psi(x\mid \hat{x})$, whereas Theorem 3.1 is stated in terms of $q\_\theta(\cdot\mid \hat{z}\_r)$.
>
> In the revision, we will define the induced surrogate conditional $\tilde q\_{\theta,\psi}(x\mid \hat z\_r):=q\_\psi\left(x\mid \Phi\_\theta^{0\leftarrow r}(\hat z\_r)\right)$, where $\hat{x}=\Phi\_\theta^{0\leftarrow r}(\hat z\_r)$ is deterministic given $\hat z\_r$. This yields $\mathbb{E}\_{p(z\_r)}\left[D\_{\mathrm{KL}}(p(x\mid z\_r)\|q\_\theta(\cdot\mid \hat z\_r))\right]
> \le\mathbb{E}\_{p(z\_r)}\left[D\_{\mathrm{KL}}(p(x\mid z\_r)\|q\_\psi(\cdot\mid \hat x))\right]+\varepsilon\_{\mathrm{link}}(\theta,\psi)$, where $\varepsilon\_{\mathrm{link}}:=\mathbb{E}\_{p(z\_r)}\mathbb{E}\_{p(x \mid z\_r)}
> \left[\log \frac{q\_\psi(x \mid \hat x)}{q\_\theta(x \mid \hat z\_r)}\right]$ is the residual linking term. Combining this with Proposition 4.4 gives the missing chain to InfoNCE up to $\varepsilon\_{\mathrm{link}}$. We will therefore soften the claim accordingly: InfoNCE is a tractable proxy for the conditional mismatch, rather than a direct bound on the original $q\_\theta$ term.
> We will make this clear in our revised version.
>
> ## W3&Q3. Contrastive mechanism vs. DINOv2 representations
> We conduct ablations to disentangle these two factors.
>
> Keeping the same pretrained representation source (**DINOv2-L**), we compare several objectives on MeanFlow-B/2 over ImageNet-256:
>
> | Method | NFE=1 | NFE=2 |
> |:-|:-:|:-:|
> | MeanFlow-B/2 (base) | 6.04 | 5.17 |
> | MeanFlow-B/2 (w/ Cos. sim.) | 5.83 | 5.09 |
> | MeanFlow-B/2 (w/ feature L2) | 5.75 | 5.06 |
> | MeanFlow-B/2 (w/ NT-Xent) | 5.72 | 5.03 |
> | MeanFlow-B/2 (w/ InfoNCE) | 5.71 | 5.02 |
>
> This shows that **DINOv2-based supervision already gives clear gains**, while contrastive objectives further improve over plain representation matching.
>
> To further isolate the role of contrastive learning from pretrained DINOv2 features, we also consider a **contrastive-without-DINO** setting, where frozen DINOv2 features are replaced by internal MeanFlow features plus a lightweight projection head:
>
> | Method | NFE=1 |
> |:-|:-:|
> | MeanFlow-XL/2 (base) | 3.43 |
> | MeanFlow-XL/2 (w/ Cos. sim.) | 3.23 |
> | MeanFlow-XL/2 (w/ feature L2) | 3.15 |
> | MeanFlow-XL/2 (w/ NT-Xent) | 3.07 |
> | MeanFlow-XL/2 (w/ InfoNCE) | 3.05 |
>
> This shows that **contrastive learning still improves the baseline even without DINOv2**, so the gain is not solely due to pretrained perceptual features. Overall, CFMM benefits from **both** strong semantic supervision and the contrastive formulation.
>
> ## W4&Q4. Extension to multi-step sampling
> Our point here is mainly practical. During training, we sample random interval pairs $(t,r)$, so the model is exposed to diverse step lengths rather than a single fixed jump. As a result, the objective improves reconstruction under a wide range of imperfect intermediate states. Since multi-step sampling is composed of such transitions, improving each step’s endpoint mapping also benefits the overall multi-step chain.
>
> ## W5&Q5. Direct comparison with REPA
> We compare CFMM with REPA on ImageNet-256 using the same FMM baseline:
>
> | Method | NFE=1 | NFE=2 |
> |:-|:-:|:-:|
> | MeanFlow-B/2 (base) | 6.04 | 5.17 |
> | MeanFlow-B/2 (+REPA) | 5.80 | 5.09 |
> | MeanFlow-B/2 (+CFMM) | 5.71 | 5.02 |
>
> Both REPA and CFMM improve the base model, while CFMM achieves better FID in both settings.
>
> ## W6. Fixed-seed qualitative comparisons
> Although the same seed does not guarantee identical outputs across different model parameters, it still provides a controlled side-by-side comparison under matched initial noise and sampling schedules. We will include such fixed-seed qualitative comparisons in the revised manuscript.

---

> > ### Author Rebuttal · Reviewer_nnvS · 2026-04-04
> >
> > I thank the authors for the additional ablations and comparisons, which are helpful. However, my main concerns remain. The theoretical contributions, i.e., originally presented as formal guarantees, have been downgraded across the board to "surrogates" and "proxies". For a paper that uses its theoretical framework as a central motivation, this retreat substantially weakens the contribution. I maintain my score.

---

> > > ### Author Response · Authors · 2026-04-06
> > >
> > > We sincerely thank the reviewer for the continued feedback and for raising this important concern.
> > >
> > > We would like to clarify that our revision does **not downgrade the theoretical contribution**, but rather **refines its interpretation** to avoid overstating what can be guaranteed in practice.
> > >
> > > In particular, Theorem 3.1 and its KL decomposition remain **unchanged and exact**.
> > > It rigorously characterizes the modeling gap as the sum of:
> > > - a marginal mismatch term, and
> > > - a conditional mismatch term.
> > >
> > > This decomposition is still the **core theoretical contribution** and the main motivation of CFMM.
> > >
> > > The change we made is purely **interpretational at the optimization level**.
> > > In practice, both components of the decomposition involve intractable quantities under flow-map sampling.
> > > Therefore, any implementable objective must necessarily rely on **tractable surrogates**. For example, the MeanFlow-style regression can be understood as a surrogate for reducing marginal mismatch, rather than an exact bound in its current form.
> > >
> > > Importantly, this does not weaken the framework; instead, it **aligns the theory with standard practice in generative modeling**, where exact objectives (e.g., KL terms) are commonly optimized via tractable surrogates (e.g., score matching, variational bounds).
> > > In our case, the decomposition serves as a **principled diagnostic tool** that identifies the two failure modes, and CFMM provides **sampling-aligned objectives** to target them in practice.
> > >
> > > We will further revise the paper to make this distinction clearer, emphasizing that:
> > > - the theoretical decomposition is exact and remains the foundation of the method, and
> > > - the proposed objectives are **principled surrogates derived from this analysis**, rather than heuristic replacements.
> > >
> > > We hope this clarification addresses the reviewer's concern.

---

### Official Review · Reviewer_hPRA · 2026-03-12

**Soundness:** 3
**Presentation:** 2
**Significance:** 3
**Originality:** 3
**Overall Recommendation:** 5
**Confidence:** 3

**Summary:**

This paper proposes Contrastive Flow Map Matching (CFMM), a training framework for flow map models (FMMs) that combines average-velocity regression with a sampling-aligned InfoNCE loss. The authors derive a reverse-KL upper bound that decomposes the model-data gap into a marginal mismatch over intermediate states and a conditional mismatch in mapping intermediates to endpoints, motivating the two objectives. CFMM is applied as a plug-in fine-tuning procedure or from-scratch training, showing consistent FID improvements on CIFAR-10, ImageNet-256, and LSUN (consistency models) with modest extra training and no inference overhead.

**Compliance With Llm Reviewing Policy:**

Affirmed.

**Final Justification:**

The authors provided thorough additional experiments and clear explanations during the rebuttal process. These additions strengthen the paper and successfully resolve my initial concerns. Therefore, I recommend acceptance.

**Key Questions For Authors:**

1. Please refer to weakness 2 about Proposition 4.2: Can you empirically correlate reductions in the MeanFlow surrogate loss with reductions in a proxy for DKL(p(zr)||qθ(ẑr)), e.g., via learned density ratios or MMD in z-space?

2. Please refer to weakness 3. I wonder if you train the critic ψ at all or if it is always a frozen DINOv2 encoder. If frozen, can you clarify how the theoretical link to the conditional KL term involving qθ(·|·) should be interpreted? Would joint training of ψ and θ make the bound tighter in practice?

3. Beyond FID, did you evaluate semantic consistency metrics (e.g., attribute preservation, object detection scores) to substantiate the claim that the contrastive term improves semantic fidelity?

**Limitations:**

No.
From my perspective, reliance on a fixed external feature encoder (DINOv2) introduces dependence on out-of-domain pretraining data and may implicitly alter the training signal in ways not captured by the bound. I would appreciate it if the author could provide an explanation and adequately discuss potential limitations.

**Strengths And Weaknesses:**

## Strength
1. The paper is well-structured. The figure and narrative around marginal vs. conditional mismatches are intuitive and help frame the method’s role. Mathematical derivations are mostly clear, with proofs deferred to the appendix.
2. The core idea is interesting. The reverse-KL decomposition into marginal and conditional terms provides a clear lens to analyze training–sampling mismatch in FMMs and motivates complementary objectives. Moreover, the framework is broadly compatible with multiple FMM variants (MeanFlow, TiM, α-Flow, Shortcut) and also applies to consistency models, indicating generality.
3. Extensive evaluations across datasets and backbones demonstrate consistent, albeit moderate, gains under 1–8 NFEs with small additional training budgets. Sensible ablations probe λ weighting, stop-gradient configurations, DINOv2 backbone variants, and batch size, strengthening the empirical case.

## Weakness
1. Limited analysis of sensitivity to the number and selection of negatives (K) in InfoNCE; for class-conditional settings, negative sampling across classes versus within-class is not clarified, which can materially affect contrastive learning. Furthermore, there is a lack of wall-clock training time and compute profiling for the added contrastive term (beyond iteration counts) and limited reporting of variance across seeds.

2. The marginal mismatch bound (Proposition 4.2 in line 240) assumes a Gaussian conditional qθ(ẑr|zt) and reduces to squared error in z-space, whereas the actual MeanFlow training optimizes a JVP-based surrogate target u_tgt rather than directly regressing to z_r. The gap between the theoretical regression objective and the practical MeanFlow loss is not fully addressed. The authors should clarify the conditions under which minimizing the MeanFlow surrogate correlates with the regression implied by the bound.

3. Some typos need to be fixed. Some notational inconsistencies (e.g., mixing z_r and ẑ_r in joint distributions within Theorem 3.1) and occasional typos may confuse readers following the derivations closely. The learnable-versus-frozen status of the critic is not consistently presented: earlier, it is described as learnable; later, DINOv2 is used frozen.

---

> ### Author Rebuttal · Authors · 2026-03-31
>
> We thank the reviewer for the thoughtful comments.
> ### W1.1. Limited analysis of sensitivity to negatives (K) in InfoNCE.
> In our implementation, negatives are drawn from the same mini-batch, so K=batch size-1.
> We therefore analyze sensitivity to K by varying the batch size:
> |Method|NFE=1|NFE=2|
> |:-|:-:|:-:|
> |MeanFlow-B/2 (base)|6.04|5.17|
> |MeanFlow-B/2 (K=255)|5.76|5.05|
> |MeanFlow-B/2 (K=511)|5.71|5.00|
> |MeanFlow-B/2 (K=1023)|5.71|5.02|
>
> The performance is stable across larger K, indicating that CFMM is not highly sensitive to the negative-pool size in practice. We will clarify this in the revision.
> ### W1.2. Lack of wall-clock training time.
> We additionally measure wall-clock time on A100 GPUs. CFMM increases training time from **1.08 s/iter** to **1.34 s/iter** for MeanFlow-B/2, and from **1.89 s/iter** to **2.37 s/iter** for MeanFlow-XL/2, i.e., about **25%** overhead. We also report FLOPs below:
> |Method|FID|FLOPs (G)|
> |:-|:-:|:-:|
> |MeanFlow-B/2|6.04|23.1|
> |MeanFlow-B/2+CFMM|5.71|34.2|
> |MeanFlow-XL/2|3.47|119.0|
> |MeanFlow-XL/2+CFMM|2.98|141.5|
>
> Thus, CFMM adds only moderate **training-time** cost, while inference remains unchanged and incurs no extra overhead.
> ### W1.3. Limited reporting of variance across seeds.
> We conducted **3 independent fine-tuning runs** with different training seeds, and for each model repeated evaluation **10 times** with different sampling seeds.
> For clarity, we provide the mean and standard deviation of the one-NFE FID on ImageNet-256 for representative baselines:
> |Method|Shortcut-XL/2|MeanFlow-B/2|MeanFlow-XL/2|TiM-XL/2|$\alpha$ -Flow-XL/2|$\alpha$ -Flow-XL/2+|
> |:-|:-:|:-:|:-:|:-:|:-:|:-:|
> |base FID|10.60|6.04|3.47|7.11|2.95|2.58|
> |Improved FID(mean $\pm$ std)|9.74 $\pm$ 0.12|5.76 $\pm$ 0.08|2.98 $\pm$ 0.09|6.87 $\pm$ 0.06|2.81 $\pm$ 0.05|2.48 $\pm$ 0.03|
>
> The standard deviations are much smaller than the corresponding gains, showing that the improvements are stable across seeds.
> ### W2&Q1. Can you empirically correlate reductions in the MeanFlow loss with reductions in a proxy for $D\_{\mathrm{KL}}(p(z\_r)\|q_\theta(\hat{z}\_r))$?
> Our intention in Proposition 4.2 is to motivate average-velocity matching as a surrogate for reducing the **marginal mismatch** over intermediate states, rather than to claim that the practical MeanFlow surrogate is itself a direct estimator of $D_{\mathrm{KL}}(p(z\_r)\|q\_\theta(\hat{z}\_r))$.
>
> To empirically examine this connection, we use **MMD in $z$-space** as a nonparametric proxy for the discrepancy between the ground-truth bridge samples $z_r$ and the model-generated intermediate samples $\hat{z}\_r$. Concretely, for matched $(x,e,t,r)$, we construct $z\_r = (1-r)x + re, \hat{z}\_r = z\_t - (t-r)u\_\theta(z\_t,r,t)$, and then compute the MMD between the empirical sets $z_r$ and $\hat{z}_r$.
> In this experiment, we fix $t=0.038$ and $r=0.022$, sample 1000 images from ImageNet-256, and evaluate the corresponding MMD values. The average MMD decreases from **132.2** for the baseline to **98.7** for CFMM, indicating that improving the surrogate is indeed correlated with better alignment of the intermediate distribution.
> This provides empirical support for the intended role of the surrogate in reducing marginal mismatch, even though the practical loss is not a direct optimization of the KL term itself.
>
> ### W3&Q2. Is $\psi$ fixed as a frozen DINOv2 encoder? If yes, how should the link to the conditional KL term be interpreted, and would joint training of $\psi$ and $\theta$ be beneficial?
> In practice, we **freeze DINOv2** and update only the generator/FMM parameters $\theta$. We will clarify this in the revision. Under this implementation, the theoretical link should be interpreted as a **surrogate connection**: the conditional KL involving $q\_\theta(\hat{x}\mid \hat{z}\_r)$ is intractable, so we introduce the auxiliary distribution $q\_\psi(x\mid \hat{x})$ and optimize it in a fixed DINOv2 feature space. Thus, the KL decomposition identifies the endpoint error, while frozen-encoder InfoNCE is the practical surrogate used in training.
> In principle, jointly training $\psi$ and $\theta$ could make the surrogate more adaptive, but it also adds optimization complexity and reduces stability. Empirically:
> |Method|FID|FLOPs (G)|
> |:-|:-:|:-:|
> |MeanFlow-B/2|6.04|23.1|
> |MeanFlow-B/2+CFMM(freeze DINOv2)|5.71|34.2|
> |MeanFlow-B/2+CFMM(joint train DINOv2)|5.73|45.6|
>
> Jointly training DINOv2 does not improve performance in our setting, while incurring substantially higher compute cost. This supports our frozen-DINOv2 design.
>
> ### Q3. Did you evaluate semantic consistency metrics.
> Our paper focuses on **unconditional generation** rather than attribute-controlled or semantic-preservation settings, so attribute- or detection-based metrics are not fully aligned with our task. As complementary evaluation beyond FID, we instead report standard generative metrics such as **Precision / Recall**, which better reflect sample quality and diversity in this setting.

---

> > ### Author Rebuttal · Reviewer_hPRA · 2026-04-01
> >
> > Thank the authors for providing the thorough additional experiments and clear explanations, especially the response to my question 1. These additions solidify the work and successfully address my initial concerns. Therefore, I will increase my rating to 5.

---

> > > ### Author Response · Authors · 2026-04-06
> > >
> > > We sincerely thank the reviewer for the positive and encouraging feedback. We are glad that the reviewer recognizes the motivation, technical contributions, and empirical effectiveness of our framework.

---

### Official Review · Reviewer_Hqrj · 2026-03-13

**Soundness:** 2
**Presentation:** 3
**Significance:** 2
**Originality:** 2
**Overall Recommendation:** 4
**Confidence:** 3

**Summary:**

This paper proposes Contrastive Flow Map Matching (CFMM), an add-on framework for flow map matching (FMM) models to recude the mismatch between training transitions and the model-induced sampling trajectory. The key idea is to complement standard average-velocity regression with a sampling-aligned contrastive objective defined on model outputs and the corresponding ground-truth targets using a pre-trained vision encoder. The method is motivated by a reverse-KL upper bound that decomposes the modeling gap into a marginal mismatch over intermediate states and a conditional mismatch in endpoint reconstruction. The resulting objective combines the original average-velocity loss with an InfoNCE loss computed in a frozen DINOv2 feature space. The method is presented as a training-only plug-in regularization loss for pre-trained FMMs and can also be used from scratch. Experiments on CIFAR-10, ImageNet 256×256, and LSUN show consistent improvements across several FMM-style baselines, without adding inference-time cost.

**Compliance With Llm Reviewing Policy:**

Affirmed.

**Final Justification:**

Considering the answers and the corresponding adjustments, my concerns have been resolved and I would like to raise my score.

**Key Questions For Authors:**

1. **Can the authors clarify more explicitly how optimizing the proposed contrastive objective is expected to reduce the original conditional mismatch term in the KL decomposition?**
   This is the main point affecting my confidence in the theoretical justification.

2. **How robust are the reported gains across random seeds, especially on the strongest ImageNet settings where the absolute FID improvements are small?**
   Reporting mean/std for a few representative cases would make the empirical claims much stronger.

3. **How should CFMM be understood relative to recent representation-alignment methods for diffusion/generative training?**
   I would like a clearer explanation of what is fundamentally new here beyond using pretrained visual features as auxiliary supervision.

4. **Did the authors test stronger nearby baselines beyond swapping InfoNCE with other distance functions?**
   A stronger representation-based baseline would help isolate the value of the specific CFMM design.

**Limitations:**

It's better to discuss the dependence on a frozen pretrained visual encoder.

**Strengths And Weaknesses:**

### Strengths

- The paper addresses a relevant problem in fast generative modeling: the mismatch between the training objective and the model-induced sampling trajectory in FMMs.

- The method is simple and practically appealing. It is a training-only modification, does not change inference, and is applicable across different backbones.

- The empirical section is fairly broad. The method is evaluated on multiple backbones and datasets, and the improvements are generally consistent, which makes the contribution more convincing than a single-model result.

- The paper is clearly written overall. The motivation is easy to follow, and the core idea can be understood without too much effort.

### Weaknesses

- My main concern is that the theory-to-method connection is not fully tight. The paper motivates the approach via a KL-based decomposition, but the actual optimization uses an auxiliary contrastive surrogate defined through frozen pretrained encoders. This is reasonable, but the paper currently presents the connection as more principled than it really is.

- The related-work positioning needs to be sharper. Using pretrained visual encoders as auxiliary supervision for generative training is already an active line of work, and the paper does not clearly explain enough what is specifically new here beyond adapting this idea to FMMs with a sample-level InfoNCE loss. Also the connection between the previous methods and CFMM is not very clear.

- Some of the gains, especially on the strongest ImageNet baselines, are relatively small. Since no seed variance or repeated-run statistics are reported, it is hard to judge how robust those improvements are.

---

> ### Author Rebuttal · Authors · 2026-03-31
>
> We thank the reviewer for the careful and constructive comments.
>
> ### Q1. How does the contrastive objective help reduce the conditional mismatch term in the KL decomposition?
> The connection is at the level of the **targeted failure mode** in Theorem 3.1: $D\_{\mathrm{KL}}(p(x,z\_r)\|q\_\theta(\hat{x},\hat{z}\_r))=D\_{\mathrm{KL}}(p(z\_r)\|q\_\theta(\hat{z}\_r))+\mathbb{E}\_{p(z\_r)}\left[D\_{\mathrm{KL}}(p(x\mid z\_r)\|q\_\theta(\hat{x}\mid \hat{z}\_r))\right]$
> This shows that fast FMM sampling is limited by both **intermediate transport mismatch** and **endpoint mismatch** from imperfect $\hat z_r$ to $\hat x$. Average-velocity matching mainly addresses the former, while our contrastive term targets the latter.
>
> Concretely, the contrastive loss supervises outputs along the model’s own path $z_t \rightarrow \hat z_r \rightarrow \hat x$, improving endpoint reconstruction under the intermediate errors encountered at test time. Since $q_\theta(\hat{x}\mid \hat{z}_r)$ is implicit and intractable, we introduce the auxiliary conditional $q\_\psi(x\mid \hat{x})
> =\frac{p\_{\mathrm{data}}(x)\exp(T\_\psi(\hat{x},x))}{Z\_\psi(\hat{x})},Z\_\psi(\hat{x})=\mathbb{E}\_{x\sim p\_{\mathrm{data}}}[\exp(T\_\psi(\hat{x},x))]$, and optimize it with InfoNCE. This yields a tractable, sampling-aligned surrogate that pulls $\hat x$ toward its matched target and away from mismatched samples, improving robustness when $\hat z\_r$ is imperfect.
>
> Thus, we do **not** claim that InfoNCE exactly minimizes the conditional KL term; rather, the KL decomposition identifies the endpoint failure mode, and InfoNCE serves as a practical surrogate.
>
> ### Q2. How robust are the gains across random seeds, especially on strong ImageNet baselines?
> To reduce randomness, we reran fine-tuning **3 times** with different training seeds. For each model, we further repeated evaluation **10 times** with different sampling seeds and report the average across all runs.
>
> For one-NFE FID on ImageNet-256, results are:
> |Method|Shortcut-XL/2|MeanFlow-B/2|MeanFlow-XL/2|TiM-XL/2|$\alpha$-Flow-XL/2|$\alpha$-Flow-XL/2+|
> |:--|:--:|:--:|:--:|:--:|:--:|:--:|
> |Base FID|10.60|6.04|3.47|7.11|2.95|2.58|
> |Improved FID|9.74 ± 0.12|5.76 ± 0.08|2.98 ± 0.09|6.87 ± 0.06|2.81 ± 0.05|2.48 ± 0.03|
>
> The gains are stable: the standard deviations are much smaller than the improvements, indicating they are not due to favorable single runs. The extra cost is modest: TiM-XL/2 uses only 35k extra iterations on top of 750k, and $\alpha$-Flow-XL/2 uses only 50k on top of 1200k, both below 4% additional training.
>
> ### Q3. How does CFMM relate to recent representation-alignment methods?
> CFMM is related because it also uses pretrained visual features as training-time supervision. However, our goal is not generic semantic regularization, but to address the **training-sampling mismatch of FMMs** in fast sampling.
>
> Imperfect finite-time transport produces a model-induced intermediate state $\hat z_r$ that deviates from the ideal bridge state $z_r$, which then degrades the final mapping to $\hat x$. This FMM-specific failure mode motivates our marginal/conditional decomposition and the resulting two-part objective.
>
> A key difference is **where the supervision is applied**: our contrastive term is imposed on **sample-level outputs along the flow-map path** $z_t \rightarrow \hat z_r \rightarrow \hat x$, rather than on hidden features alone. It is paired with average-velocity matching because the latter improves intermediate transport, while the former strengthens endpoint reconstruction under model-induced errors.
>
> Hence, CFMM should be viewed as a **sampling-aligned endpoint objective for FMMs**, not a generic representation regularizer. It is training-only, adds no inference-time overhead, and can also be used from scratch. We will revise the related-work discussion to clarify this positioning.
>
> ### Q4. Did you test stronger nearby baselines beyond swapping InfoNCE with other distance functions?
> Yes. Using the same pretrained representation source (**DINOv2-L**), we compared several **representation-based objectives** beyond InfoNCE, including cosine similarity, feature-space L2 regression, and NT-Xent. This isolates whether the gain comes from the specific CFMM design or from strong semantic supervision.
>
> Using MeanFlow-B/2 on ImageNet-256, the FID scores are:
>
> |Method|NFE=1|NFE=2|
> |:--|:--:|:--:|
> |Base|6.04|5.17|
> |+ Cos. sim.|5.83|5.09|
> |+ feature L2|5.75|5.06|
> |+ NT-Xent|5.72|5.03|
> |+ InfoNCE|5.71|5.02|
>
> All representation-based objectives improve over the baseline, while InfoNCE performs best among these nearby alternatives. NT-Xent is closely related to InfoNCE, which likely explains their similar performance. In addition, Table 6 compares InfoNCE with L1, L2, LPIPS, and Pseudo-Huber; these are generally weaker than the DINOv2-based objectives, suggesting that the main gain comes from semantically meaningful representation supervision rather than merely changing the metric form.

---

> > ### Author Rebuttal · Reviewer_Hqrj · 2026-04-04
> >
> > I thank the authors for the corresponding answers and explanations. The paper is a useful resource for the training-time refinement of the FMM method using pre-trained representation sources. My concerns have been resolved and I would like to raise my score to 4.

---

> > > ### Author Response · Authors · 2026-04-06
> > >
> > > We sincerely thank the reviewer for the careful and insightful comments.
> > > We greatly appreciate the constructive feedback and the recognition of our work.

---

### Decision · Program_Chairs · 2026-04-30

**Decision:**

Accept (regular)

**Comment:**

The paper makes a clear practical contribution: CFMM is a simple training-only refinement for flow map models that yields consistent gains across multiple baselines and datasets, with no inference-time overhead. The rebuttal also strengthened the empirical case by adding seed statistics, compute overhead, stronger nearby baselines, and a direct comparison to REPA.

The main remaining weakness is the mismatch between the paper’s theoretical framing and what is currently established. The core decomposition in Theorem 3.1 is useful, but the stronger claims linking the training objectives to the original KL terms are not fully justified in the current draft. In rebuttal, the authors acknowledge that Proposition 4.2 is too strong as written and that the InfoNCE connection should be interpreted as a tractable surrogate rather than a direct bound on the original conditional mismatch. There also remains a reasonable concern that part of the empirical gain may come from strong representation supervision rather than the specifically contrastive component.

Overall, I recommend **acceptance**, but I strongly encourage a more careful and substantially expanded empirical study in the final version. In particular, the paper would benefit from stronger isolation of the specific contribution of the contrastive objective relative to feature/perceptual supervision, as well as clearer empirical support for the proposed interpretation. The paper should also narrow its theoretical claims, clearly distinguish exact results from surrogate arguments, and moderate novelty claims tied to the theoretical justification.